# Hierarchical Information Aggregation for Incomplete Multimodal Alzheimer's Disease Diagnosis

**Chengliang Liu[1,2,3]**, **Yuanxi Que[1]**, **Qihao Xu[3]**, **Yabo Liu[4]**,
**Jie Wen[3]**, **Jinghua Wang[3]**, **Xiaoling Luo[1]***

[1]College of Computer Science and Software Engineering, Shenzhen University
[2]Laboratory for Artificial Intelligence in Design, The Hong Kong Polytechnic University
[3]School of Computer Science and Technology, Harbin Institute of Technology, Shenzhen
[4]College of Artificial Intelligence, Ocean University of China
liucl1996@163.com, {queyuanxi, xqh51199597, xiaolingluoo}@outlook.com,
yaboliu.ug@gmail.com, jiewen_pr@126.com, wangjh2012@foxmail.com

## Abstract

Alzheimer's Disease (AD) poses a significant health threat to the aging population, underscoring the critical need for early diagnosis to delay disease progression and improve patient quality of life. Recent advances in heterogeneous multimodal artificial intelligence (AI) have facilitated comprehensive joint diagnosis, yet practical clinical scenarios frequently encounter incomplete modalities due to factors like high acquisition costs or radiation risks. Moreover, traditional convolution-based architecture face inherent limitations in capturing long-range dependencies and handling heterogeneous medical data efficiently. To address these challenges, in our proposed heterogeneous multimodal diagnostic framework (HAD), we develop a multi-view Hilbert curve-based Mamba block and a hierarchical spatial feature extraction module to simultaneously capture local spatial features and global dependencies, effectively alleviating spatial discontinuities introduced by voxel serialization. Furthermore, to balance semantic consistency and modal specificity, we build a unified mutual information learning objective in the heterogeneous multimodal embedding space, which maintains effective learning of modality-specific information to avoid modality collapse caused by model preference. Extensive experiments demonstrate that our HAD significantly outperforms state-of-the-art methods in various modality-missing scenarios, providing an efficient and reliable solution for early-stage AD diagnosis.

## 1 Introduction

AD is a progressive neurodegenerative disorder characterized by cognitive impairment, gradual loss of memory, and a decline in self-care abilities as its primary clinical manifestations [1, 2, 3]. Due to the lack of effective cures, AD poses a significant threat to the health of the elderly population, severely impacting the quality of life of patients and their families and imposing a heavy medical burden on society [4]. Mild Cognitive Impairment (MCI) is considered a precursor stage of AD, marked by mild cognitive decline without a noticeable impact on daily functional abilities. Early diagnosis and intervention during this stage are critical for delaying disease progression and improving patients' quality of life. In recent years, with the rapid increase of the type of multimodal data, researchers have been able to better understand and diagnose early-stage AD from multiple perspectives, providing more comprehensive and objective decision-making support for clinical diagnosis and treatment [5, 6].

---

*Corresponding author: Xiaoling Luo (email: xiaolingluoo@outlook.com).

39th Conference on Neural Information Processing Systems (NeurIPS 2025).

These advances also lay the groundwork for the application of multimodal Artificial Intelligence (AI) in joint AD diagnosis.

However, in practical clinical settings, the collection of multimodal data is often hampered by issues such as radiation risks, high costs, and unexpected patient withdrawal, leading to the frequent problem of missing modalities. This makes it difficult to obtain complete multimodal datasets [7, 8]. To address missing data, existing studies often discard cases with incomplete modalities and rely solely on data with complete modalities for analysis [9]. This approach reduces the subject scale, thereby limiting the performance of models. To overcome this issue, researchers have proposed various multimodal learning frameworks based on strategies such as subspace learning, knowledge distillation, and missing data imputation [10, 11]. For instance, data imputation methods use generative models like Generative Adversarial Networks (GANs) and Autoencoders to fill in missing modalities, thereby expanding the training dataset. However, due to challenges in ensuring the quality of imputed data, such methods often introduce redundant or even misleading information, resulting in decreased performance. Furthermore, the heterogeneity of multimodal data in AD, which ranges from three-dimensional (3D) image data to 1D biomarker data, presents additional challenges. Effectively integrating heterogeneous multimodal data and mining their shared semantic information under conditions of missing modalities remains a key difficulty in AI-assisted AD diagnosis.

Currently, mainstream multimodal diagnostic methods are typically based on 3D Convolutional Neural Networks (CNNs) or Transformer architectures [12, 13]. However, 3D CNNs often struggle to effectively capture long-range spatial dependencies and are constrained by large parameter sizes and high computational costs [14]. While Transformer-based models can model long-range dependencies, their computational complexity increases quadratically with the input data dimensions. This is particularly problematic when dealing with high-dimensional multimodal 3D medical image data, where computational inefficiency becomes a significant bottleneck, severely limiting their practical clinical applications. Recently, Structured State Space Models (SSMs), exemplified by Mamba, have gained attention for their efficient information extraction capabilities and linear computational complexity. For example, Liu et al. [15] proposed VMamba with a 2D Selective Scan module (SS2D) that bridges the ordered nature of 1D selective scan and the non-sequential structure of 2D visual data. Xing et al. [16] developed SegMamba, which captures remote dependencies in the entire 3D voxel at multi-scale. In the context of 3D multimodal medical image, exploring an efficient feature extraction module based on SSMs holds the potential to enhance diagnostic performance while reducing model complexity, thereby better meeting the demands of clinical applications.

To address the aforementioned challenges, this paper proposes a novel heterogeneous multimodal diagnostic framework for AD, named HAD. The framework is capable of processing multimodal data that includes Magnetic Resonance Imaging (MRI), Positron Emission Tomography (PET), Cerebrospinal Fluid (CSF), and Clinical Assessment Data (CAD) with arbitrary modality missing, providing a flexible solution for early-stage AD diagnosis. One the one hand, to address the challenges of long-range dependencies and high computational complexity in 3D brain image data, we develop a hierarchical spatial feature extraction module. Building upon existing work [17], we adopt the same Hilbert curves for space-filling transformations while maintaining their core concept of "locality-preserving property of space-filling curves". However, our proposed HSFE module introduces two key innovations: (1) A hierarchical architecture based on the fractal theory of 3D Hilbert curves enables multi-scale information fusion through multi-level recursive structures; (2) A multi-directional scanning mechanism (incorporating axial rotation and mirror transformations) enhances complementary spatial information capture. This module integrates traditional convolutional operation with efficient state space model, enabling the effective capture of both shallow features and global dependencies in 3D image data. On the other hand, to enhance the consistency of discriminative capabilities across different modalities for AD diagnosis, we propose an optimization objective based on maximizing mutual information between multimodal joint semantic features and shallow features in the semantic embedding space. This strategy ensures that the framework effectively integrates heterogeneous information from various modalities, improving diagnostic performance in scenarios with incomplete multimodal datasets. Our main contributions are summarized as follows:

- We propose a heterogeneous multimodal AD diagnosis framework capable of handling arbitrary incomplete modalities. Unlike existing methods primarily tailored for 3D brain image data, our framework can effectively process heterogeneous multimodal data with significant informational and structural differences.

- To effectively handle complex 3D MRI and PET image data, we propose the multi-view Hilbert curve-based Mamba block (HMamba), along with a hierarchical spatial feature extraction strategy built upon HMamba. These modules alleviate discontinuities introduced by spatial voxel serialization and unify long-sequence modeling with local feature extraction across multiple scales.

- We propose a multimodal semantic representation learning framework, which establishes a goal of maximizing mutual information between modality-specific features and semantic labels, simultaneously considering modality-specific information extraction and consistency representation learning.

## 2 Preliminary

### 2.1 Problem Definition

Given a multimodal AD diagnosis dataset $\mathcal{D} = (\mathbf{x}, \mathbf{y})$ with $n$ subjects, and each subject $x$ consists of heterogeneous multimodal data (e.g., structural MRI, PET, CSF, and CAD), denoted as $x = \{x^{(1)}, x^{(2)}, \ldots, x^{(m)}\}$, where $m$ denotes the total number of modalities. Noted that we use $x^{img}$ to indicate the MRI or PET data. The corresponding diagnostic label is denoted as $y$, representing the clinical status of the subject (e.g., cognitively normal (CN), MCI, and AD). In practical clinical scenarios, some modalities might be missing for certain subjects due to various reasons, thus we let $\mathcal{V}$ denote the set of available modalities and $|\mathcal{V}| \leq m$. The objective of multimodal AD diagnosis is thus to train a neural network model capable of accurately predicting the clinical label $\mathbf{y}$ using any available subset of modalities, even when some modalities are missing during inference.

### 2.2 State Space Modals

State Space Models (SSMs) are classical linear time invariant systems widely used in control theory and signal processing, characterized by their linear complexity and effectiveness in modeling sequential data. Recently, SSMs have gained renewed attention in deep learning due to their ability to efficiently capture long-range dependencies. Gu et al. [18] first introduced the HiPPO framework, providing a theoretically optimal approach to represent continuous-time state-space models by high-order polynomial projections. Subsequently, Gu et al. proposed the Structured State Space Sequence model (S4) [19] that introduces discretization and convolutional representation for parallel training, demonstrating superior performance in processing time series data. More recently, Mamba [20] introduced selective structured state spaces, simplifying the architecture and improving parallelism. Various vision Mamba architectures [15, 21] further extended the state-space modeling paradigm from sequences to two-dimensional image data, effectively addressing the quadratic complexity issue inherent in vision Transformers.

### 2.2.1 Hilbert Curve for 3D Brain Image Data

Existing Vision Mamba methods typically convert structured 2D or 3D visual data into 1D sequences through serialization approaches such as bi-directional scanning, cross scanning, or continuous scanning [22, 23, 24]. However, these simple scanning strategies inevitably disrupt the inherent spatial relationships, causing spatially adjacent pixels or voxels to become distant from each other in the serialized sequence. Such spatial discontinuity significantly impairs the model's ability to capture local structural information and long-range spatial dependencies, thereby limiting diagnostic performance in medical imaging tasks. To alleviate this issue, we propose adopting the 3D Hilbert curve for scanning brain image data. Unlike traditional scanning approaches [25, 26, 27], the Hilbert curve is a continuous, space-filling fractal curve known for its excellent locality-preserving property . Specifically, the Hilbert curve mapping ensures that voxels spatially close in the 3D space remain close in the serialized 1D representation to a large extent, thus minimizing spatial distortion and aiding the model in effectively capturing local and global contextual information from structured medical image data.

The 3D Hilbert curve can be generated iteratively, with each iteration referred to as the curve's *order* $N$. At each iteration, the curve recursively subdivides the original 3D cubic space into $2^N \times 2^N \times 2^N$ sub-cubes, as shown in Fig. 5. The Hilbert curve of order $N$ is thus constructed by connecting the Hilbert curves of order $(N-1)$ from eight smaller sub-cubes through rotations and reflections,

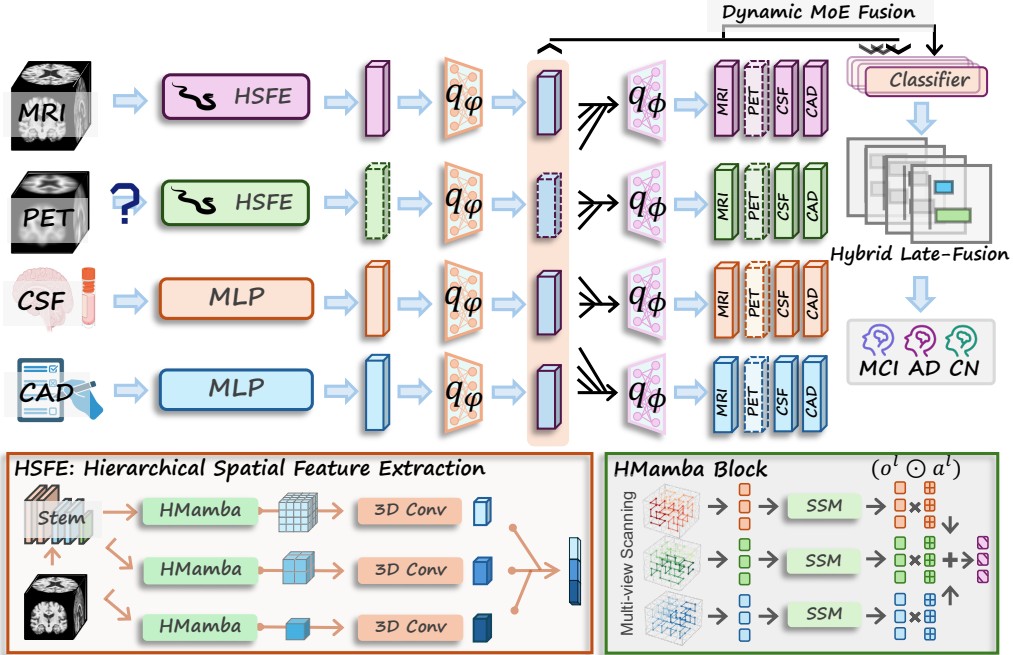

Figure 1: The schematic diagram of our HAD. It contains 2 main parts, a heterogeneous modality-specific shallow feature extraction consisting of HSFE module and MLP (left half) and high-level multimodal semantic coding (right half). "$q_\varphi$" and "$q_\phi$" denote the modality-specific encoder and cross-modal decoder, respectively; "3D Conv" denotes the 3D residual convolution module.

preserving the locality and continuity of the space-filling curve. Formally, given a voxel coordinate $(x_c, y_c, z_c)$ within a cubic voxel grid of size $2^N$, the 3D Hilbert curve defines a mapping $\mathcal{H}_N(\cdot)$ from the 3D coordinate to a 1D sequence index $h$:

$$h = \mathcal{H}_N(x_c, y_c, z_c), \quad h \in \{0, 1, \ldots, 8^N - 1\}. \tag{1}$$

An appropriate Hilbert curve order $N$ is selected based on the input image data resolution.

## 3 Method

### 3.1 HMamba: Hilbert Curve-Based Mamba Block

**Multi-View Spatial Scanning.** As illustrated in Fig. 5, spatially adjacent voxels located at the boundaries between neighboring sub-cubes might become relatively distant within the serialized sequence due to the intrinsic fractal structure of the Hilbert curve. This spatial tearing phenomenon potentially leads to the loss of critical adjacency information, adversely affecting the model's ability to capture fine-grained spatial dependencies. To bridge this gap and further alleviate the spatial discontinuity issue, we propose utilizing multiple Hilbert curves oriented along different spatial axes, complementing the conventional approach of scanning with a Hilbert curve along a single orientation. In other words, we expect that

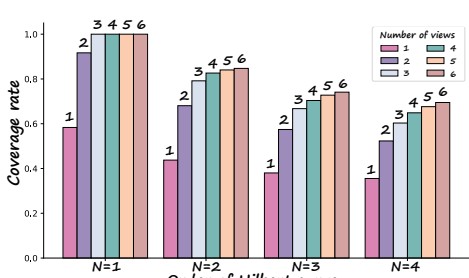

Figure 2: Coverage rate vs. order of Hilbert curve under different view numbers.

multiple Hilbert curves collectively maximize the coverage of adjacency edges among voxels, effectively preserving comprehensive spatial context within serialized data. A higher coverage rate indicates better preservation of spatial adjacency information and thus facilitates more effective modeling of spatial context (please refer to Appendix A.3 for the definition of *coverage rate*).

As we know, starting from a fixed vertex, six independent curves can be drawn. Therefore, in Fig. 2, we illustrate the relationship between the number of scanning views and the coverage rate. At lower orders, three-view scanning is sufficient to fully cover all adjacent edges. However, as image resolution increases, achieving complete coverage becomes more challenging. Furthermore, increasing the number of scanning curves should be considered with caution due to the additional computational overhead. Overall, for input $x^{img} \in \mathbb{R}^{d_i \times d_i \times d_i}$, $L$ multi-view scanning layers are denoted as $\{\mathcal{F}_S^l : x^{img} \in \mathbb{R}^{d_i \times d_i \times d_i} \to \mathbb{R}^{d_i^3}\}_{l=1}^L$.

**Multi-View Dynamic Fusion-Based Mamba Block.** Upon serializing the input 3D image data, we employ the SSMs to capture long-range dependencies across the resulting long sequences, which originate from continuous-time linear dynamical systems, mapping an input $x_t \in \mathbb{R}$ to an output $o_t \in \mathbb{R}$ via a hidden state $h_t \in \mathbb{R}^{d_e}$ as follows:

$$h_t = Ah_{t-1} + Bx_t, \quad o_t = Ch_t, \tag{2}$$

where $A \in \mathbb{R}^{d_e \times d_e}$, $B \in \mathbb{R}^{d_e \times 1}$, and $C \in \mathbb{R}^{1 \times d_e}$ are learnable parameters.

For discrete sequence modeling, the continuous parameters $(A, B)$ are discretized using zero-order hold (ZOH) with a step size $\Delta$: $\overline{A} = e^{\Delta A}$, $\overline{B} = A^{-1}(e^{\Delta A} - I)B$. The resulting discrete-time SSM is formulated as:

$$h_t = \overline{A}h_{t-1} + \overline{B}x_t, \quad o_t = Ch_t. \tag{3}$$

In practice, model outputs are efficiently computed through convolution:

$$o = x * H, \quad H = (C\overline{B}, C\overline{AB}, \ldots, C\overline{A}^{M-1}\overline{B}), \tag{4}$$

where $M = d_i^3$ is the input sequence length, and $H \in \mathbb{R}^M$ is the structured convolution kernel. Since sequences derived from different views correspond to spatially misaligned voxel indices, a reverse indexing operation is subsequently applied to map the multi-view serialized features back to the original 3D voxel grid structure, i.e., $\bar{\mathcal{F}}_S^l : o \in \mathbb{R}^{d_i^3} \to \mathbb{R}^{d_i \times d_i \times d_i}$. Furthermore, due to the distinct scanning views, the SSMs applied to each serialized sequence capture complementary spatial dependencies. To effectively aggregate these multi-view features, we propose a voxel-wise dynamic fusion strategy. Formally, given the encoded feature tensors from $L$ distinct scanning views, denoted as $\{o^l\}_{l=1}^L$, we assign a learnable spatially adaptive weighting tensor $a^l \in \mathbb{R}^{d_i \times d_i \times d_i}$ for each view. Each element of $a^l$ dynamically balances the importance of each voxel from the $l$-th view. The fused voxel-wise feature map $\hat{o}$ is thus obtained as a weighted combination: $\hat{o} = \sum_{l=1}^L a^l \odot o^l$, where $\odot$ denotes the voxel-wise (element-wise) multiplication. The weighting tensors $a^l$ are optimized during training, enabling the model to dynamically emphasize the most informative features from different scanning views for each voxel individually. This proposed voxel-wise dynamic multi-view fusion effectively integrates complementary spatial context captured by multi-view SSMs, significantly enhancing the model's representation capability for 3D medical image data.

### 3.2 Hierarchical Spatial Feature Extraction

As described in Section 3.1, our proposed multi-view Hilbert curve-based scanning approach significantly mitigates the spatial tearing issue. However, at the high imaging resolutions (i.e., larger Hilbert curve orders), the multi-view scanning strategy may fail to resolve all spatial discontinuities, as complete adjacency edge coverage becomes increasingly challenging. Motivated by the hierarchical receptive field scaling property inherent in multi-scale CNNs, we propose a hierarchical spatial feature extraction (HSFE) module that serializes and processes 3D image data at multi-scale resolutions. At each hierarchical level, the resolution of the input 3D image data is progressively reduced by the downsampling operation. Specifically, given an original 3D image with dimensions $2^N \times 2^N \times 2^N$, we iteratively construct $K$ lower-resolution images by a series of downsampling modules consisting of convolutional layer and MaxPool layer: $\{\hat{x}|_{k+1} = \text{Down}(\hat{x}|_k)\}_{k=0}^{K-1}$, where $\hat{x}|_0 = \text{Stem}(x^{img})$ and Stem module is to expand channels and reduce dimensions. At $k$-th level, we utilize HMamba module with $N - k$ order Hilbert curve and 3D residual convolution block to model long-sequence dependency and local spatial information:

$$g|_k = \mathcal{C}^k(\mathcal{M}^k(\hat{x}|_k)), \quad k \in [0, K-1], \tag{5}$$

where $g|_k$ denotes the $k$-th level output. $\mathcal{C}^k$ and $\mathcal{M}^k$ mean the corresponding 3D residual block and HMamba module, respectively. Then, we simply concatenate all $K$ outputs and perform max-pooling

to obtain the final output of the HSFE module:

$$g = \text{MaxPool}(\text{Concat}(g|_0, g|_1, \ldots, g|_{K-1})). \tag{6}$$

This hierarchical strategy naturally alleviates the spatial discontinuity issues that may arise from high-order Hilbert curves, as the receptive field expands significantly at deeper layers. Consequently, our hierarchical state space architecture effectively compresses spatial features at various scales, progressively enhancing the information density of the learned representations.

### 3.3 Multimodal Semantic Representation Learning

Heterogeneous multimodal data inherently exhibit a modality gap, with different modalities contributing distinct perspectives to AD diagnosis. To preserve modality-specific characteristics, we avoid inter-modality information interactions in the initial stage instead of conducting modality-specific feature extraction (i.e., HSFE module for 3D image data and Multi-Layer Perceptrons (MLPs) for CSF and CAD). This is to map the heterogeneous multimodal data into a unified feature space, facilitating further cross-modal alignment [28, 29]. In general, we aim for the learned features to preserve semantic information as fully as possible, while simultaneously striving to achieve semantic consistency across multiple modalities. Therefore, we propose the following mutual information maximization objective:

$$\max I(\mathbf{g}; \mathbf{y}) + \alpha I(\mathbf{g}; \mathbf{z}), \tag{7}$$

where $\mathbf{g} = \{\mathbf{g}^{(v)}\}_{v \in \mathcal{V}}$ and random variable $\mathbf{g}^{(v)}$ corresponds to the modality-specific feature of $v$-th modality. $\mathbf{z}$ represents the cross-modal joint semantic representation and $\alpha$ is the balanced parameter. Eq. (7) consists of two parts: the first term focus on learning discriminative information from labels, while the second term aims to align modality-specific representations with cross-modal semantic representations. For the first term of Eq. (7), the equivalent objective is as follows:

$$\max I(\mathbf{g}; \mathbf{y}) \Leftrightarrow \min H(P, Q), \tag{8}$$

where $P \sim p(\mathbf{y}|\mathbf{g})$ and $Q \sim q(\mathbf{y}|\mathbf{g})$ denote the real distribution of $\mathbf{y}$ and predicted distribution, respectively, and $H(P, Q)$ is the cross entropy. Specifically, we employ a dynamic Mixture-of-Experts (MoE) fusion to get the typical joint posterior of latent representation $\mathbf{z}$, i.e., $q_\varphi(\mathbf{z}|\mathbf{g}) = \frac{1}{|\mathcal{V}|} \sum_{v \in \mathcal{V}} \omega^{(v)} q_{\varphi_v}(\mathbf{z}|\mathbf{g}^{(v)})$, where $\omega^{(v)} = \frac{e^{\eta^v/\tau}}{\sum_{v \in \mathcal{V}} e^{\eta^v/\tau}}$ ($\tau$: the temperature parameter) is calculated by the modality-specific learnable parameters $\{\eta^1, \eta^2, ..., \eta^m\}$, and then inference the prediction probability by parameterized neural networks $q_\theta(\mathbf{y}|\mathbf{z})$. This corresponds to constructing the probabilistic graph model: $\mathbf{g} \to \mathbf{z} \to \mathbf{y}$. Together with the second term, this approach simultaneously ensures semantic consistency across multimodal representations and facilitates semantic learning of disease categories. However, it ignores the inherent heterogeneity among different modalities, which can hinder the effective exploration of multimodal complementary information. In addition, due to discrepancy in information and data structure, the network commonly has obvious fitting preference for certain modalities, which can easily lead to training collapse issue for hard-fitting modalities (see further analysis in Section 4.3). Thus, we propose to add a direct inference of $\mathbf{y}$ from modality observations, i.e., $\mathbf{g} \to \mathbf{y}$. To be specific, we introduce the modality-specific conditional distribution into the final prediction distribution as follows:

$$q(\mathbf{y}|\mathbf{g}) =: \frac{1}{2} q_\theta(\mathbf{y}|\mathbf{z}) + \frac{1}{2} \sum_{v \in \mathcal{V}} \omega^{(v)} q_{\theta_v}(\mathbf{y}|\mathbf{g}^{(v)}), s.t., \sum_{v \in \mathcal{V}} \omega^{(v)} = 1, \tag{9}$$

Finally, given the established prediction distribution in Eq. (9), cross entropy minimization objective given in Eq. (8) can be expressed as the cross entropy loss function $\mathcal{L}_{ce} = \text{CrossEntropy}(y, \hat{y})$, where $\hat{y}$ denotes the joint prediction probability. By introducing dynamic weighting factors, the importance across different modalities can be effectively balanced. Note that we reverse $\omega^{(v)} = \frac{e^{-\eta^v/\tau}}{\sum_{v \in \mathcal{V}} e^{-\eta^v/\tau}}$ during the training stage to effectively mitigate the insufficient modal fitting issue caused by the model's learning preference toward certain modalities. Formally, Eq. (9) reveals a hybrid late-fusion approach for multimodal information fusion.

For the second term of Eq. (7), we can get the following lower bound:

$$I(\mathbf{g}; \mathbf{z}) \geq \mathbb{E}_{\mathbf{g} \sim p(\mathbf{g})} \left[ \int p(\mathbf{z}|\mathbf{g}) \log q_\phi(\mathbf{g}|\mathbf{z}) d\mathbf{z} \right], \tag{10}$$

where $q_\phi(\mathbf{g}|\mathbf{z})$ is a variational approximation to the true posterior $p(\mathbf{g}|\mathbf{z})$. Based on the conditional independence assumption across modalities [30, 31], and multimodal MoE fusion strategy [32], the lower bound can be further simplified and rewritten as:

$$\mathbb{E}_{\mathbf{g}\sim p(\mathbf{g})}\left[\int p(\mathbf{z}|\mathbf{g})\log q_{\phi_v}(\mathbf{g}|\mathbf{z})d\mathbf{z}\right] = \frac{1}{|\mathcal{V}|}\sum_{v\in\mathcal{V}}\mathbb{E}_{\mathbf{g}^{(v)}\sim p(\mathbf{g}^{(v)})}\left[\int p(\mathbf{z}|\mathbf{g}^{(v)})\log q_{\phi_v}(\mathbf{g}^{(v)}|\mathbf{z})d\mathbf{z}\right]$$
$$+ \frac{1}{|\mathcal{V}|}\sum_{v,u\in\mathcal{V},v\neq u}\mathbb{E}_{\mathbf{g}^{(v)}\sim p(\mathbf{g}^{(v)})}\left[\int p(\mathbf{z}|\mathbf{g}^{(v)})\log q_{\phi_v}(\mathbf{g}^{(u)}|\mathbf{z})d\mathbf{z}\right].$$

(11)

Therefore, the goal of $\max I(\mathbf{g};\mathbf{z})$ is transformed into minimizing the reconstruction loss $\mathcal{L}_{intra}$ and $\mathcal{L}_{inter}$, corresponding to the first (intra-modal reconstruction) and second terms (inter-modal reconstruction) of Eq. (11), respectively. Our overall loss function is $\mathcal{L} = \mathcal{L}_{ce} + \lambda\mathcal{L}_{intra} + \gamma\mathcal{L}_{inter}$, where $\lambda$ and $\gamma$ are the penalty parameters replacing $\alpha$ in Eq. (7).

## 4 Experiments

### 4.1 Experimental settings

**Dataset**. The data utilized in this study is collected from the Alzheimer's Disease Neuroimaging Initiative (ADNI), a publicly available database designed to facilitate research into biomarkers and clinical trials for AD. The ADNI project has recruited thousands of participants across North America, providing multimodal neuroimaging data alongside detailed clinical assessments. Specifically, we select baseline T1-weighted structural MRI and paired $^{18}$F-AV45 PET images as bi-modal brain imaging; we collect the values of biomarkers, such as amyloid $\beta$-protein (A$\beta$), Tau, and p-Tau, as CSF data; and 29 clinical cognitive examination scores as the CAD. All data is from four ADNI subsets: ADNI-1, ADNI-2, ADNI-3, and ADNI-GO. Subjects are categorized into three diagnostic groups: CN, MCI, and AD. Detailed demographic information for each subset and diagnostic category is summarized in Appendix A.4.

**Preprocessing**. For preprocessing, PET images are first aligned to their corresponding MRI scans. Subsequently, both MRI and PET images are spatially normalized to the standard Montreal Neurological Institute (MNI) space using Statistical Parametric Mapping (SPM) [33]. Intensity normalization and Gaussian smoothing are also applied to PET images to reduce image noise and standardize intensity values. Finally, skull-stripping procedure is conducted on both MRI and PET images using FreeSurfer [34] to remove non-brain tissues and further enhance data quality for subsequent analysis.

We collect a total of 2345 subjects with complete MRI, PET, and CAD modalities, that is, the above three modalities for each subject are available. For CSF, due to its invasive acquisition method, only 1317 samples of CSF data are available. Furthermore, in order to simulate different modality missing situations, we randomly mask [10%, 30%, 50%] instances of MRI, PET, and CAD modalities by filling in 0 value at the missing position, while ensuring that at least one of the modalities is available for each subject. Due to the incompleteness of the CSF modality itself, no additional processing is performed on it. Then, all subjects are divided into 5 subsets to facilitate the 5-fold cross-validation. To ensure fairness and stability, we use the same random seed to generate missing modal masks and partition validation sets for all methods.

**Competitor and Evaluation Metric**. In this study, we compare our proposed method with several state-of-the-art incomplete multimodal learning frameworks, i.e., LMVCAT [35], Adapted [36], DMRNet [37], ShaSpec [38], GMD [39], CM3T [40], and TriMF [41]. Most methods are difficult to directly apply to our heterogeneous multimodal data due to differences in downstream tasks or designs. Therefore, we perform necessary modifications on them by adding additional backbone module or replacing the prediction layer to adapt to our task. Following previous studies [42, 43, 44], we evaluate the effectiveness of our method using five metrics: the area under the ROC curve (AUC), accuracy (ACC), F1-score (F1), sensitivity (SEN), and specificity (SPE). Higher values of these metrics indicate better performance.

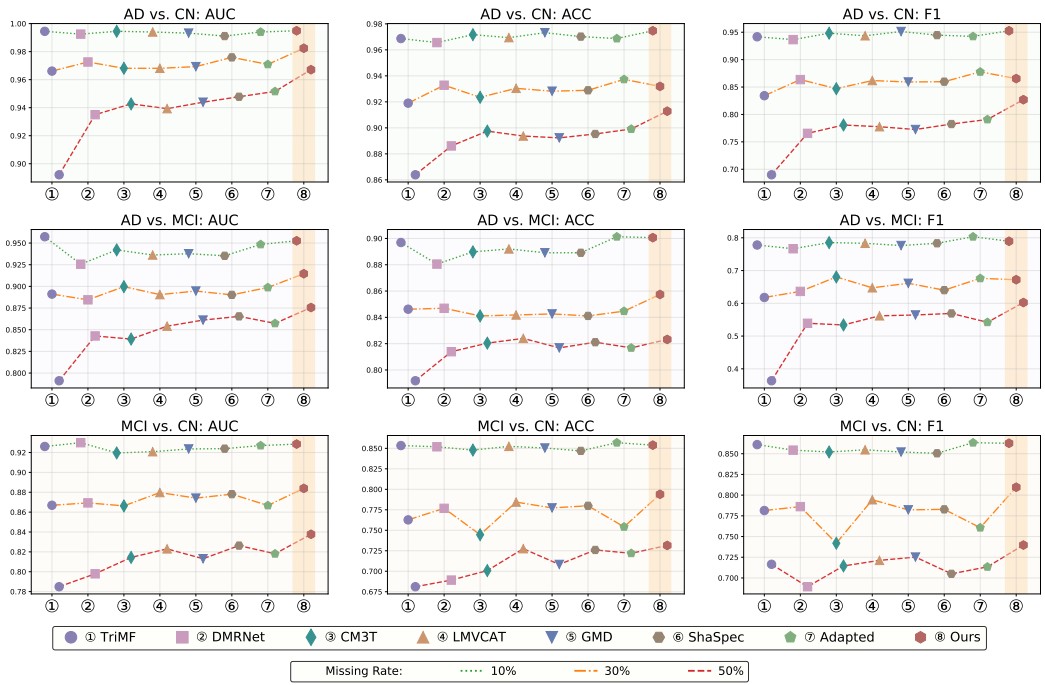

Figure 3: The comparison results of eight methods on three tasks with different missing rates.

## 4.2 Experimental Analysis

To investigate the performance of our HAD, following most existing methods [2, 3, 9], we conduct experiments on three tasks (AD vs. CN, AD vs. MCI, and MCI vs. CN) using five-fold cross-validation. Our HAD is compared with seven state-of-the-art methods under various modality missing rates as shown in Fig. 3. From Fig. 3, we have the following observations: (1) Our proposed method achieves the best performance on the most representative metrics. Specifically, although SEN and SPE can often exhibit an imbalance—one metric being very high and the other very low due to their definitions in binary classification problems—our method still demonstrates superior performance when considering these two metrics jointly; (2) Comparing the three binary classification tasks, it is evident that all methods exhibit the highest discriminative capability in distinguishing AD from CN, and the lowest in distinguishing MCI from CN. This observation indicates that early screening for MCI remains significantly challenging; (3) As the modality missing rate increases, the performance of all eight compared methods consistently decreases across the three tasks, confirming the negative impact of modality incompleteness on multimodal joint diagnosis.

## 4.3 Modality Imbalance Study

As discussed above, the pronounced heterogeneity of our AD multimodal data means that training all inputs uniformly can bias the model toward specific modalities. To assess whether the proposed composite late-fusion strategy mitigates this imbalance, we perform a controlled study on the AD-versus-CN task under a 50% missing rate. Fig. 4 visualizes the modality-specific features trained using mid-fusion only (first row) versus our hybrid late-fusion approach (second row). As shown in Fig. 4 (a)-(d), conventional mid-fusion leads to insufficient training of MRI and PET modality-specific features (exhibiting poor class discriminability) due to the rapid convergence of encoders on CSF and CAD modalities. In contrast, Fig. 4 (e)-(h) demonstrate that our hybrid late-fusion approach effectively mitigates this issue, enabling balanced training of all modality-specific encoders, particularly for MRI and PET. We attribute this phenomenon to the inherent heterogeneity in multimodal data. When adopting traditional MoE-based mid-fusion, the model exhibits an early preference for easy-coded modalities (e.g., CSF and CAD), consequently neglecting the training of the MRI and PET branches. Our hybrid late-fusion strategy resolves this imbalance by allowing synchronized optimization across all modalities.

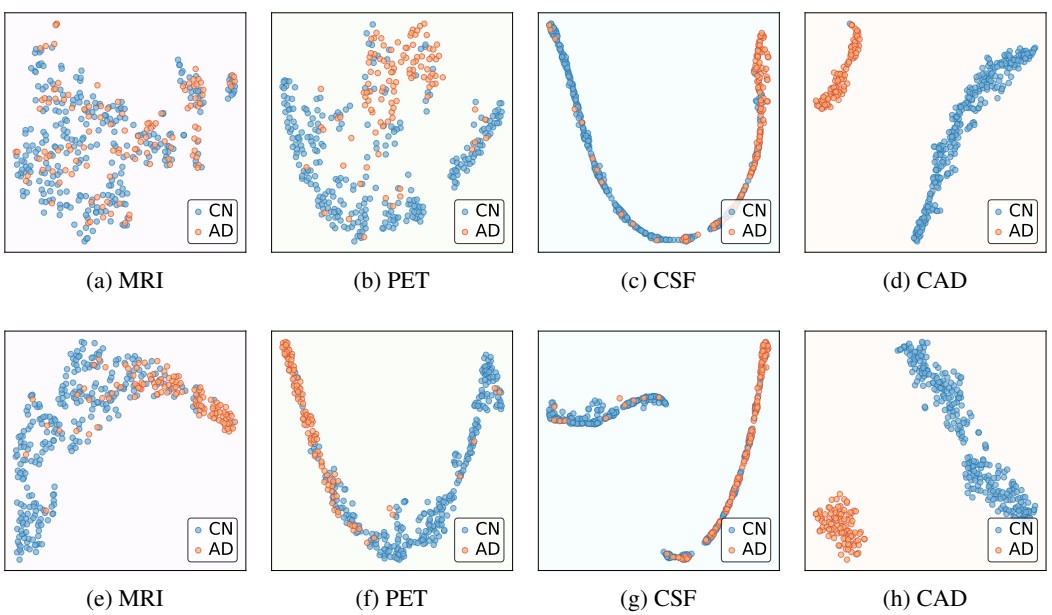

Figure 4: T-SNE visualization of modality-specific features at 20th training epoch. Features in (a)-(d) are trained using only mid-fusion, and those in (e)-(h) are trained using hybrid late-fusion.

## 4.4 Ablation Study

To further investigate the effectiveness of each design component within our HAD, we conduct ablation experiments in this section. Firstly, we ablate individual terms from our total loss function by removing parameters $\beta$ and $\gamma$ separately, and evaluate the performance on the MCI vs. CN task with 50% modality availability. From Table 1, it can be observed that both intra-modal and inter-modal reconstruction losses contribute positively to the

Table 1: Ablation study on MCI vs. CN task under 50% missing rate.

| Method | AUC | ACC | F1 | SEN | SPE |
|---|---|---|---|---|---|
| HAD *w/o* $\mathcal{L}_{inter}$ | 0.812 | 0.717 | 0.719 | 0.699 | 0.742 |
| HAD *w/o* $\mathcal{L}_{intra}$ | 0.810 | 0.705 | 0.724 | 0.759 | 0.641 |
| HAD *w/o* $\mathcal{L}_{inter}$ and $\mathcal{L}_{intra}$ | 0.805 | 0.712 | 0.725 | 0.739 | 0.680 |
| HAD *w/o late-fusion* | 0.810 | 0.738 | 0.741 | 0.722 | 0.760 |
| HAD *w/o DF* | 0.805 | 0.710 | 0.704 | 0.666 | 0.765 |
| HAD *w single-HSFE* | 0.814 | 0.726 | 0.734 | 0.728 | 0.732 |
| HAD *w/o HSFE* | 0.813 | 0.710 | 0.706 | 0.675 | 0.757 |
| HSFE *w/o HMamba* | 0.816 | 0.713 | 0.707 | 0.690 | 0.738 |
| **HAD** | **0.838** | **0.732** | **0.740** | **0.740** | **0.735** |
| HMamba *w SegMamba* | 0.805 | 0.713 | 0.733 | 0.756 | 0.670 |
| HMamba *w VMamba* | 0.800 | 0.722 | 0.723 | 0.720 | 0.725 |
| HMamba *w/o CA* | 0.816 | 0.722 | 0.742 | 0.770 | 0.685 |

model performance. Specifically, intra-modal reconstruction aims at compressing intra-modal information, thereby preserving all modality-specific information. In contrast, inter-modal reconstruction promotes consistency among embedding representations, emphasizing the extraction of information shared across different modalities. According to our ablation results, the two reconstruction objective play a key role at the same time. To study the effectiveness of the hybrid late-fusion strategy, the direct inference from modality-specific information is deleted, i.e., converting Eq. (9) to $q(\mathbf{y}|\mathbf{g}) = q_\theta(\mathbf{y}|\mathbf{z})$, denoting as "HAD *w/o late-fusion*". Then, we perform ablation study on the dynamic factors in the dynamic MoE fusion strategy (let $\omega^{(v)} = \frac{1}{|\mathcal{V}|}$), representing as "HAD *w/o DF*". Finally, we replace the HMamba block inside HSFE with a 3D ResNet50 module ("HSFE *w/o HMamba*"), so that the hierarchical structure is preserved but the long-range modeling of HMamba is removed. From Table 1, we find that hybrid late-fusion based on multiple predictions brings significant performance improvements. In addition, the dynamic learnable parameters have a positive impact on both the mid-fusion and late-fusion process.

Next, regarding the HSFE module designed for 3D imaging data, we first remove the entire HSFE structure and use a vanilla 3D ResNet50 as the backbone for the image modality, denoted as "HAD *w/o HSFE*", and simplify the structure by removing the hierarchical design, denoted as "HAD *w single-HSFE*". Furthermore, to validate the effectiveness of our multi-view Hilbert curve-based scanning strategy within the HMamba module, we remove the cross-view attention mechanism (setting $a^l = 1$), denoted as "HMamba *w/o CA*", and replace our proposed multi-view spatial scanning approach with alternative scanning strategies (e.g., three-axis scanning [16] and cross scanning [15]), denoted respectively as "HMamba *w SegMamba*" and "HMamba *w VMamba*". Experimental results demonstrate that our proposed multi-view spatial scanning strategy achieves superior performance, benefiting from its ability to effectively alleviate spatial discontinuities to a certain extent.

## 4.5 Conclusion

In this paper, we propose HAD, a heterogeneous multimodal diagnostic framework that effectively addresses core challenges in multimodal AD diagnosis, such as modality heterogeneity and modality incompleteness. Through the innovative design of our multi-view Hilbert curve-based HMamba module and HSFE, the proposed model effectively captures long-range spatial dependencies in 3D medical images. Moreover, our multimodal semantic representation learning framework, leveraging intra- and cross-modal reconstruction, significantly enhances semantic consistency across heterogeneous modalities and remain the modal-specific complementary information [45]. Comprehensive experimental evaluations confirm that our HAD consistently achieves significant performance advantages under various modality-missing cases. Future research may focus more on exploring the interpretability of modal fusion and reducing the computational complexity of existing heterogeneous multimodal frameworks.

## Acknowledgments

We thank Yuanxi Que for leading the programing and experimental validation of this work, and for the original insight of our proposed HMamba backbone. We thank Prof. Waikeung Wong for substantial suggestions of ideas and important assistance during the rebuttal period. We are also grateful to Dr. Zhuangzhuang Li for support with data collection. Finally, we thank all reviewers and ACs for their thorough and constructive feedback.

This work was supported by the National Natural Science Foundation of China [No. 62502320], the Natural Science Foundation of Guangdong Province [No. 2025A1515010184, 2024A1515011816], the project of Shenzhen Science and Technology Innovation Committee [No. JCYJ20240813141424032], the Foundation for Young Innovative Talents in Ordinary Universities of Guangdong [No. 2024KQNCX042], and the Scientific Foundation for Youth Scholars of Shenzhen University [No. 827-0001083].

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

## A  Appendix

### A.1  Key Derivation Procedure

In this subsection, we give the key derivation procedure of multimodal semantic learning object $I(\mathbf{g}; \mathbf{z})$:

$$
\begin{aligned}
&I(\mathbf{g}; \mathbf{z}) \\
&= \int \int p(\mathbf{g}, \mathbf{z}) \log \frac{p(\mathbf{g}|\mathbf{z})}{p(\mathbf{g})} d\mathbf{g} d\mathbf{z} \\
&\geq \int p(\mathbf{g}) \int p(\mathbf{z}|\mathbf{g}) \log p(\mathbf{g}|\mathbf{z}) d\mathbf{g} d\mathbf{z} \\
&= \int p(\mathbf{g}) \int p(\mathbf{z}|\mathbf{g}) \log q_\phi(\mathbf{g}|\mathbf{z}) d\mathbf{g} d\mathbf{z} + \\
&\quad \int p(\mathbf{g}) \int p(\mathbf{z}|\mathbf{g}) \log \frac{p(\mathbf{g}|\mathbf{z})}{q_\phi(\mathbf{g}|\mathbf{z})} d\mathbf{g} d\mathbf{z} \\
&\geq \mathbb{E}_{\mathbf{g} \sim p(\mathbf{g})} \left[ \int p(\mathbf{z}|\mathbf{g}) \log q_\phi(\mathbf{g}|\mathbf{z}) d\mathbf{z} \right].
\end{aligned}
\tag{12}
$$

For $p(\mathbf{z}|\mathbf{g})$, we adopt the dynamic MoE fusion strategy to model $p(\mathbf{z}|\mathbf{g})$:

$$p(\mathbf{z}|\mathbf{g}) = \frac{1}{|\mathcal{V}|} \sum_{v \in \mathcal{V}} p(\mathbf{z}|\mathbf{g}^{(v)}). \tag{13}$$

Based on multimodal conditional independence, we have:

$$q_\phi(\mathbf{g}|\mathbf{z}) = \prod_{v \in \mathcal{V}} q_{\phi_v}(\mathbf{g}^{(v)}|\mathbf{z}). \tag{14}$$

Combined Eq. (12), Eq. (13), and Eq. (14), we have:

$$
\begin{aligned}
&\mathbb{E}_{\mathbf{g} \sim p(\mathbf{g})}\big[\int p(\mathbf{z}|\mathbf{g}) \log q_\phi(\mathbf{g}|\mathbf{z})d\mathbf{z}\big]\\
=&\mathbb{E}_{\mathbf{g} \sim p(\mathbf{g})}\big[\int p(\mathbf{z}|\mathbf{g}) \log \prod_{v \in \mathcal{V}} q_{\phi_v}(\mathbf{g}^{(v)}|\mathbf{z})d\mathbf{z}\big]\\
=&\mathbb{E}_{\mathbf{g} \sim p(\mathbf{g})}\big[\int \big(\frac{1}{|\mathcal{V}|} \sum_{v \in \mathcal{V}} \omega^{(v)} p(\mathbf{z}|\mathbf{g}^{(v)})\big) \log \prod_{v \in \mathcal{V}} q_{\phi_v}(\mathbf{g}^{(v)}|\mathbf{z})d\mathbf{z}\big]\\
=&\frac{1}{|\mathcal{V}|} \sum_{v \in \mathcal{V}} \mathbb{E}_{\mathbf{g}^{(v)} \sim p(\mathbf{g}^{(v)})}\big[\int p(\mathbf{z}|\mathbf{g}^{(v)}) \log q_{\phi_v}(\mathbf{g}^{(v)}|\mathbf{z})d\mathbf{z}\big]\\
&+\frac{1}{|\mathcal{V}|} \sum_{v,u \in \mathcal{V}, u \neq v} \mathbb{E}_{\mathbf{g}^{(v)} \sim p(\mathbf{g}^{(v)})}\big[\int p(\mathbf{z}|\mathbf{g}^{(v)}) \log q_{\phi_v}(\mathbf{g}^{(u)}|\mathbf{z})d\mathbf{z}\big].
\end{aligned}
\tag{15}
$$

### A.2  Multi-View Hilbert Curves

Fig. 5 shows the 3D Hilbert curves with orders $N = 1$ and $N = 2$. Fig. 6 shows the multi-view Hilbert curve-based scanning with different orders. Note that in the implementation, we adopt a bidirectional scanning mechanism in both forward and reverse directions for each curve.

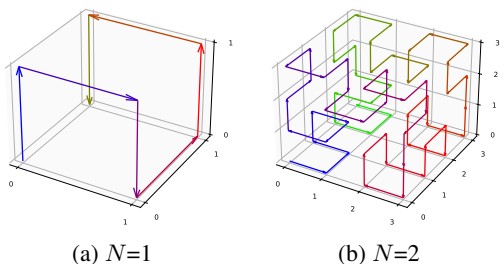

(a) $N$=1        (b) $N$=2

Figure 5: Schematic of 3D Hilbert space-filling curves with different orders.

### A.3  Definition of Coverage Rate

Formally, we quantify the effectiveness of the multi-view Hilbert scanning strategy using the concept of *coverage rate* $c$. Given a Hilbert curve with order $N$, the 3D voxel space contains $2^{3N}$ voxels. Each voxel connects with its immediate neighbors, resulting in totally $E_N = 3 \times 2^{2N}(2^N - 1)$ adjacency edges along three axes. For $L$ distinct Hilbert curves, let $\mathcal{H}_l$ denote the set containing traversed edges by $l$-th Hilbert curve, then the set of unique adjacency edges traversed by $L$ curves is defined as $\mathcal{U} = \bigcup_{l=1}^{L} \mathcal{H}_l$. Thus, the coverage rate is defined as $c = \frac{|\mathcal{U}|}{E_N}$.

### A.4  ADNI Dataset Statistical Information

In Table 2, we list the detailed information of the ADNI dataset used in our experiments.

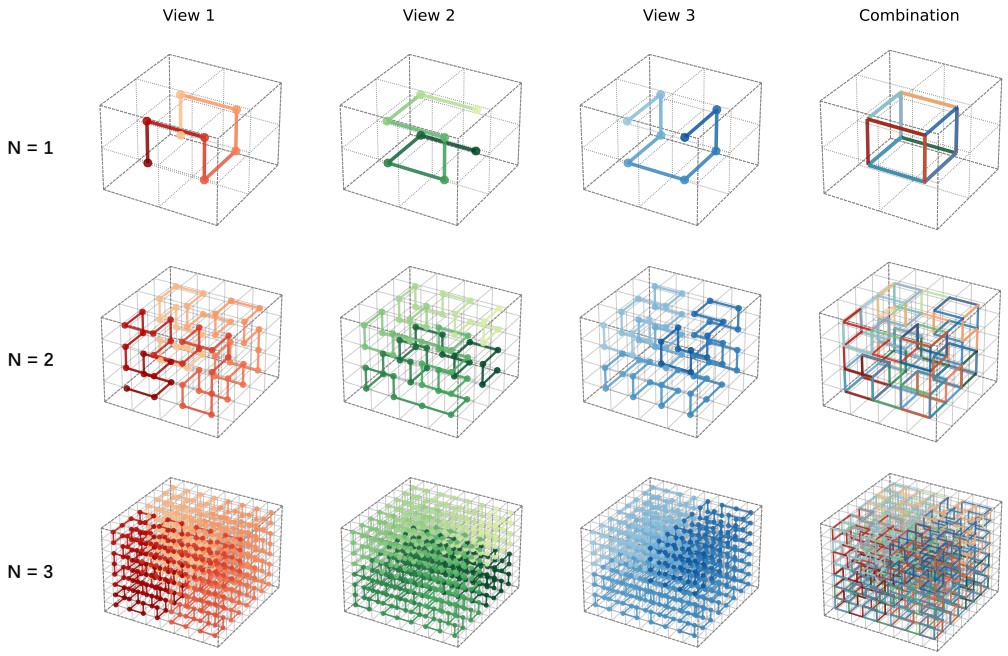

Figure 6: Schematic diagrams of multi-view Hilbert curve-based scanning with different orders.

Table 2: Demographic characteristics of subjects used in this study.

| Dataset | Category | No. of subjects | Male/Female | Age (mean ± std) |
|---------|----------|-----------------|-------------|------------------|
| ADNI-1 | CN | 159 | 87/72 | 75 ± 5 |
| | MCI | 125 | 84/41 | 74 ± 7 |
| | AD | 87 | 43/44 | 72 ± 7 |
| ADNI-2 | CN | 678 | 315/363 | 72 ± 6 |
| | MCI | 616 | 347/269 | 71 ± 7 |
| | AD | 242 | 143/109 | 74 ± 8 |
| ADNI-3 | CN | 84 | 30/54 | 70 ± 6 |
| | MCI | 37 | 24/13 | 73 ± 9 |
| | AD | 7 | 6/1 | 79 ± 5 |
| ADNI-GO | CN | 27 | 8/19 | 65 ± 5 |
| | MCI | 259 | 140/119 | 71 ± 8 |
| | AD | 14 | 9/5 | 73 ± 6 |

