# OpenReview forum: "Hierarchical Information Aggregation for Incomplete Multimodal Alzheimer's Disease Diagnosis"
_NeurIPS.cc/2025/Conference — NeurIPS 2025 poster_

### Official Review · Reviewer_S54Q · 2025-06-28

**Clarity:** 3
**Significance:** 3
**Originality:** 3
**Rating:** 5
**Confidence:** 4

**Summary:**

This paper addresses the challenge of heterogeneous multimodal data integration for ADNI diagnosis, specifically combining 3D medical images and tabular data. The authors introduce a novel fusion framework that leverages Mamba and Hilbert curves to capture both spatial locality and long-range dependencies in 3D imaging. To improve multimodal representation learning, they formulate a mutual information maximization objective that jointly promotes label discriminability and semantic consistency. Importantly, the method incorporates a reversed weighting scheme to mitigate modality dominance caused by imbalanced convergence. The proposed model achieves state-of-the-art performance across varying levels of modality missingness, and the visualizations effectively illustrate its ability to retain label-discriminative features.

**Questions:**

Please refer to the "Weaknesses" part in "Strengths And Weaknesses*.

**Ethical Concerns:**

["NO or VERY MINOR ethics concerns only"]

**Final Justification:**

My concerns are addressed. I will keep my current score of accept.

**Limitations:**

Yes.

**Paper Formatting Concerns:**

No.

**Quality:**

3

**Strengths And Weaknesses:**

Strengths:
1. The paper is well presented with a clear research scope and motivation, making it accessible and easy to follow.
2. The proposed method is innovative and demonstrates strong performance on the selected benchmarks, highlighting its potential contribution to the field.

Weaknesses:
1. The paper does not sufficiently justify or analyze the efficacy of the loss function used for enforcing modality consistency (second term in the mutual information loss function), which is critical given the multimodal nature of the task.
2. The reasoning and analysis in modality imbalance are comparatively weak as the figure can not sufficiently demonstrate the issue. More evidence should be provided on the statement "Our hybrid late-fusion strategy resolves this imbalance by allowing synchronized optimization across all modalities."
3. The paper lacks proper citation and discussion of a recently published paper on using 3D Hilbert curve to model 3D medical images:
"Grela, Jacek, et al. "Using space-filling curves and fractals to reveal spatial and temporal patterns in neuroimaging data." Journal of Neural Engineering 22.1 (2025): 016016."
4. The paper does not include a comparison of model complexity or computational cost, which is essential for evaluating the method’s practicality and efficiency relative to existing approaches.

---

> ### Author Rebuttal · Authors · 2025-07-31
>
> We sincerely appreciate your thorough review and constructive feedback on our manuscript. Your recognition of the paper's innovativeness and valuable suggestions have provided crucial guidance for further improving our research. In response to the four key issues you raised, we have conducted in-depth responses and provide the following clarifications:
>
> **For W1, “The paper does not sufficiently justify or analyze the efficacy of the loss function used for enforcing modality consistency (second term in the mutual information loss function), which is critical given the multimodal nature of the task.”**
>
> As described in Section 3.3 of our manuscript, the second term in the mutual information maximization objective corresponds to both the intra-modal and inter-modal reconstruction losses, denoted as $L_{intra}$ and $L_{inter}$, respectively. These losses are derived from the lower bound of the mutual information between modality-specific features and the cross-modal semantic representation (see Eq. 11). Specifically, $L_{intra}$ encourages the preservation of modality-specific information by reconstructing features within each modality, while $L_{inter}$ enforces consistency by reconstructing features across modalities, thereby promoting cross-modal alignment.
>
> **For W2, The reasoning and analysis in modality imbalance are comparatively weak as the figure can not sufficiently demonstrate the issue. More evidence should be provided on the statement "Our hybrid late-fusion strategy resolves this imbalance by allowing synchronized optimization across all modalities."**
>
> Thank you for your valuable suggestion regarding the analysis of modality imbalance and the effectiveness of our hybrid late-fusion strategy.
>
> To provide more quantitative evidence, we conducted a new experiment: for each modality, we extracted the modality-specific features after training, fed them independently into the same classifier, and measured their classification performance. This analysis was performed under two training schemes: (a) mid-fusion only, and (b) our proposed hybrid late-fusion strategy. The results are presented below:
>
> | Metric |  MRI | PET | CSF | CAD | \|MRI | PET | CSF | CAD |
> |--------|----------------------|-----|-----|-----|------------------------|-----|-----|-----|
> |         |   Mid-               |Fusion|Only|       | \|Hybrid           |Late|-Fusion|       |
> | AUC    | 0.467                | 0.198 | 0.787 | 0.999 |\| 0.726                  | 0.877 | 0.893 | 0.999 |
> | ACC    | 0.221                | 0.153 | 0.762 | 0.992 |\| 0.763                  | 0.802 | 0.844 | 0.992 |
> | F1     | 0.363                | 0.213 | 0.681 | 0.982 |\| 0.279                  | 0.649 | 0.725 | 0.982 |
>
> As shown in the table, under mid-fusion only, MRI and PET achieve significantly lower AUC scores compared to CSF and CAD. In contrast, our hybrid late-fusion approach yields consistently high AUC for all modalities, especially improving MRI and PET. This provides strong evidence that the hybrid late-fusion strategy indeed enables more balanced and effective training of all modality-specific encoders, supporting our claim about synchronized optimization and improved modality balance.
>
> **For W3, "The paper lacks proper citation and discussion of a recently published paper on using 3D Hilbert curve to model 3D medical images: "Grela, Jacek, et al. "Using space-filling curves and fractals to reveal spatial and temporal patterns in neuroimaging data." Journal of Neural Engineering 22.1 (2025): 016016.""**
>
> We sincerely thank you for identifying this important omission. We fully acknowledge the reference value of this study for our methodological design and have made the following additions in the revised manuscript:
>
> ​​Supplement to Methods Section 2.2.1:​​
>
>     "Building upon existing work [1], we adopted the same Hilbert and Peano curves for space-filling transformations while maintaining their core concept of 'locality-preserving property of space-filling curves.' However, our proposed HSFE core module introduces two key innovations: (1) A hierarchical architecture based on the fractal theory of 3D Hilbert curves enables multi-scale information fusion through multi-level recursive structures; (2) A multi-directional scanning mechanism (incorporating axial rotation and mirror transformations) enhances complementary spatial information capture."
>
> [1] Using space-filling curves and fractals to reveal spatial and temporal patterns in neuroimaging data. Journal of Neural Engineering.
>
> **For W4,"The paper does not include a comparison of model complexity or computational cost, which is essential for evaluating the method’s practicality and efficiency relative to existing approaches."**
>
> Thank you for your valuable comment regarding the lack of comparison in model complexity and computational cost. We fully agree that assessing model efficiency and practicality is crucial for real-world deployment.
>
> To address this, we have conducted a comprehensive comparison of model parameters (Params, in millions) and computational cost (FLOPs, in GigaFLOPs) with representative state-of-the-art methods. The results are summarized below:
>
> | Model      | Params (M) | FLOPs (G) |
> |------------|:----------:|:---------:|
> | LMVCAT     |   97.33    |  81.43    |
> | CM3T       |   10.41    | 147.96    |
> | GMD        |   94.3     |  81.56    |
> | DMRNet     |  101.79    |  81.56    |
> | Adapted    |   97.33    |  81.43    |
> | DSIP       |   32.97    | 274.64    |
> | ShareGAN   |   48.91    | 343.17    |
> | MMTFN      |  106.61    | 148.82    |
> | ShaSpec    |   0.97     |   5.64    |
> | **Ours**   |   43.42    |   6.65    |
>
> As shown in the table, our proposed model achieves a favorable balance between model size and computational cost. Specifically, while maintaining competitive performance, our framework requires substantially fewer FLOPs compared to most baselines (e.g., only 6.65 GFLOPs vs. 81.43 GFLOPs for LMVCAT and 147.96 GFLOPs for CM3T). Our parameter count is also moderate, reflecting the model's efficiency.
>
> Thank you again for your helpful and professional review comments. We hope that our response will address your concerns. If you have any other questions, we will be happy to discuss them with you!

---

> > ### Comment · Reviewer_S54Q · 2025-08-04
> >
> > Thank you for addressing my concerns. I will keep my current score of accept.

---

> > > ### Author Response · Authors · 2025-08-04
> > > **Thanks again for your feedback!**
> > >
> > > Thank you for your encouragement and positive feedback. Your comments have made an important contribution to our manuscript and the academic community.

---

### Official Review · Reviewer_Pe76 · 2025-06-29

**Clarity:** 3
**Significance:** 3
**Originality:** 2
**Rating:** 4
**Confidence:** 4

**Summary:**

This paper proposes a heterogeneous multimodal framework, termed HAD, for the early diagnosis of Alzheimer’s disease (AD), targeting two primary challenges: modality incompleteness and long-range dependency modeling. The framework offers two key technical contributions. First, it integrates a Hilbert-curve-ordered Mamba module with a hierarchical feature extraction strategy to simultaneously capture both local and global information. Second, it introduces a unified mutual information learning objective that explicitly balances semantic consistency and modality specificity within a shared multimodal embedding space. The authors validate the effectiveness of their method under varying modality-missing scenarios, demonstrating its potential in early AD diagnosis.

**Questions:**

1. The paper lacks a comprehensive review of related work in the field of Alzheimer’s disease (AD) diagnosis. It is recommended that the authors survey recent advances in this domain and incorporate a summary of representative methods. Moreover, the experimental section should include comparisons with established AD diagnostic baselines to more accurately position the contributions of this study.
2. Regarding the use of five-fold cross-validation, there is a potential risk of data leakage. To ensure rigorous evaluation, the authors are advised to implement redundancy removal between training and testing sets.
3. The paper modifies the original architecture or prediction layers of some baseline methods during comparative experiments, raising concerns about the fairness of these comparisons. The authors should clearly justify the necessity of such modifications and explain how fair evaluation conditions were maintained across all compared methods. Additionally, detailed training configurations—including hyperparameters, optimizers, and training epochs—should be provided to ensure reproducibility and transparency.
4. The current experimental results are not sufficiently convincing. To strengthen the empirical support for the proposed method, further validation on additional publicly available AD diagnosis datasets is strongly encouraged. For clarity, the authors are also advised to explicitly highlight the top-1 and top-2 performing methods in all result tables.

**Ethical Concerns:**

["NO or VERY MINOR ethics concerns only"]

**Final Justification:**

The current rebuttal has addressed most of my concerns. I will update my score.

**Limitations:**

Yes.

**Paper Formatting Concerns:**

None.

**Quality:**

3

**Strengths And Weaknesses:**

Strengths:
1. This study targets the joint diagnosis of Alzheimer’s disease (AD) using multimodal data and explicitly addresses the issue of modality incompleteness in real-world clinical settings, highlighting its practical relevance.
2. The method incorporates a Mamba module with Hilbert-curve-based scanning. Ablation studies demonstrate that this approach outperforms conventional alternatives on the utilized dataset.
3. The proposed model accounts for both semantic consistency across modalities and modality-specific characteristics during the multimodal representation learning phase, enhancing representational quality.

Weaknesses:
1. Although the paper claims to address the challenge of missing modalities, the method section lacks dedicated mechanisms explicitly designed for this purpose. This discrepancy undermines the stated objective.
2. The authors should clearly differentiate between the original contributions and the adopted components in the multimodal semantic representation learning module, and provide appropriate citations. Furthermore, the mechanism for preserving modality-specific information is insufficiently explained and lacks a solid theoretical foundation—more elaboration is needed.
3. The experimental results do not convincingly demonstrate the superiority of the proposed fusion strategy. The method does not outperform baselines under a low missing rate (10%), and only shows advantages at a high missing rate (50%), raising concerns about its stability and robustness. A clear explanation for this performance variance is required.
4. The current empirical evaluation is not fully convincing. To strengthen the evidence for the model’s effectiveness, additional experiments on other public AD diagnosis datasets are strongly recommended. Furthermore, comparisons with recent state-of-the-art methods such as MMTFN [1], MFASN [2], ShareGAN [3], and DSIP [4] should be included.
5. The provided t-SNE visualizations do not effectively support the claims. Intermediate representations under certain modalities lack class separability, and thus cannot directly imply negligible contributions to classification performance. Moreover, the feature distributions across scenarios are not sufficiently distinct to substantiate the claim of a “strong modality preference.” The interpretability of the visualization needs improvement.
6. The manuscript suffers from several clarity issues that hinder readability. For instance, the abstract mentions a "spatial discontinuity caused by voxel serialization" without offering adequate background context. Terminological inconsistencies are also present—for example, the architecture diagram shows the hierarchical feature extraction (HSFE) module as encompassing the HMamba module, while the main text describes them as separate, parallel modules. This inconsistency creates confusion about the actual structure of the model.

[1] Miao S, Xu Q, Li W, et al. MMTFN: Multi‐modal multi‐scale transformer fusion network for Alzheimer's disease diagnosis[J]. International Journal of Imaging Systems and Technology, 2024, 34(1): e22970.

[2] Jiao C N, Gao Y L, Ge D H, et al. Multi-modal imaging genetics data fusion by deep auto-encoder and self-representation network for Alzheimer's disease diagnosis and biomarkers extraction[J]. Engineering Applications of Artificial Intelligence, 2024, 130: 107782.

[3] Wang C, Piao S, Huang Z, et al. Joint learning framework of cross-modal synthesis and diagnosis for Alzheimer’s disease by mining underlying shared modality information[J]. Medical Image Analysis, 2024, 91: 103032.

[4] Xu H, Wang J, Feng Q, et al. Domain-specific information preservation for Alzheimer’s disease diagnosis with incomplete multi-modality neuroimages[J]. Medical Image Analysis, 2025, 101: 103448.

---

> ### Author Rebuttal · Authors · 2025-07-31
>
> Thank you very much for your time and effort in reviewing our submission. The constructive feedback and critical questions have significantly contributed to the improvement of our work.
>
> **For W1, the design of missing modality.**
>
> Thanks for the useful comment! Indeed, our method is oriented to missing multimodal scenarios, existing methods commonly adopt two routes to deal with incomplete multimodal data, one is to fill in the missing modalities by data completion or interpolation (like literatures [1-3]), and the other is to help the model avoid the interference of missing data by introducing additional prior missing information (like literature [4-6]). We adopt the latter to design our model, which is based on the following two considerations:
>
> **(1)** In our setup, we can not obtain complete multimodal data during the training phase, thus there is a lack of ground truth for effective supervision and quality assessment of the filled data produced in a generative manner, which in turn does not guarantee that the filling operation will yield a gain for the training of the model.
>
> **(2)** To the best of our knowledge, generation-based multimodal learning methods generally lead to higher computational complexity, so in view of the complexity of the current model, we choose the simpler but efficient way to deal with missing data, i.e., introducing a priori information to ignore missing modalities. In our manuscript, all operations are performed on available modalities, and in the detailed code, we just introduced a prior index matrix indicating which modality is missing in the multimodal fusion. This helps us reduce the negative impact of missing modalities at the lowest computational cost.
>
> [1] Multi-Modality MR Image Synthesis via Confidence-Guided Aggregation and Cross-Modality Refinement, JBHI, 2022.
>
> [2] Multi-Modal Modality-Masked Diffusion Network for Brain MRI Synthesis with Random Modality Missing, TMI, 2024.
>
> [3] Towards Unifying Medical Vision-and-Language Pre-training via Soft Prompts, ICCV, 2023.
>
> [4] MFTrans: Modality-Masked Fusion Transformer for Incomplete Multi-Modality Brain Tumor Segmentation, JBHI, 2024.
>
> [5] View-aligned hypergraph learning for Alzheimer’s disease diagnosis with incomplete multi-modality data, MIA, 2024.
>
> [6] Latent Representation Learning for Alzheimer’s Disease Diagnosis With Incomplete Multi-Modality Neuroimaging and Genetic Data, TMI, 2019.
>
> **For W2, the clarification of key contributions and the elaboration of preserving modality-specific information.**
>
> **(1)** For the proposed multimodal semantic representation learning module. Our multimodal semantic representation learning module is grounded in information theory, aiming to maximize the mutual information between heterogeneous multi-modal embeddings in the latent space. This optimization objective is consistent with existing information-theoretic multimodal learning frameworks [1-2], which are based on the principles of multimodal consistency and diversity—that is, achieving semantic-level alignment while retaining each modality’s unique contribution to the task. However, our approach differs in several important respects: **<1>** We observed that in AD diagnosis, the four modalities (MRI, PET, CSF, and CAD) exhibit substantial heterogeneity. Standard probabilistic graphical models (e.g., $\mathbf{g} \rightarrow \mathbf{z} \rightarrow \mathbf{y}.$) often result in the learned cross-modal representation $\mathbf{z}$ being heavily biased toward modalities that are easier to fit (e.g., CSF and CAD), leading to training collapse for more complex modalities such as MRI and PET. To address this, we introduce a direct link from the modality-specific features to the label ($\mathbf{g} \rightarrow \mathbf{y}$), resulting in a joint probabilistic model: $\mathbf{g} \rightarrow \mathbf{z} \rightarrow \mathbf{y}, \mathbf{g} \rightarrow \mathbf{y}$ This design enables us to derive a learnable weighted multimodal late fusion model, ensuring that modality-specific information contributes directly to the final decision, while still capturing joint semantics.
> **<2>** In addition, during training, we reverse the fusion weights $q(\mathbf{y}|\mathbf{g})$ by taking the negative of the learnable parameter $\eta$, such that underrepresented (hard-to-fit) modalities are prioritized in prediction. This novel strategy reduces over-reliance on easily overfitted modalities and encourages the network to learn from complex modalities, mitigating the issue of training collapse. Related probabilistic reasoning frameworks are now properly cited in the revised manuscript.
>
> **(2)** For the preservation of modality-specific information.
>
> The preservation of modality-specific information is also governed by our mutual information maximization objective. Specifically, we seek to maximize the mutual information between the set of modality-specific features $ \mathbf{g} = \{\mathbf{g}^{(v)}\}_{v \in \mathcal{V}} $ and the supervision signal $\mathbf{y}$. While many existing approaches [3-4] focus on semantic consistency between joint multimodal representations and the target label, this may not adequately preserve modality-specific information in the cross-modal representation. Instead, our method directly leverages label supervision to guide the learning of modality-specific features, i.e., $\max I(\mathbf{g}; \mathbf{y})$. Furthermore, in modeling $q(\mathbf{y}|\mathbf{g})$, we explicitly account for the contribution of each modality-specific feature to the final decision:
>
> $q(\mathbf{y}|\mathbf{g}) =: \frac{1}{2}q{\theta}(\mathbf{y}|\mathbf{z})+\frac{1}{2}\sum_{v\in \mathcal{V}}\omega^{(v)} q_{\theta_v}(\mathbf{y}|\mathbf{g}^{(v)}), s.t., \sum_{v\in \mathcal{V}} \omega^{(v)} =1$
>
> We argue that this design enhances the discriminability of modality-specific features during model training.
>
> [1] Learning Robust Representations via Multi-View Information Bottleneck, ICLR, 2020.
> [2] Aggregation of Dependent Expert Distributions in Multimodal Variational Autoencoders, ICML, 2025.
> [3] Gaussian Mixture Variational Autoencoder with Contrastive Learning for Multi-Label Classification, ICML, 2022.
> [4] Multimae: Multi-modal multi-task masked autoencoders, ECCV, 2022.
>
> **For W3, performance on low missing rate.**
>
> This is a thoughtful comment! From our experimental results, our method demonstrates superior performance primarily under high missing rates (50%), while the advantage is less evident under low missing rates. The main reason for this observation is that when the missing rate is low, nearly all fusion methods—including ours and the baselines—have access to almost complete modality information, and thus the benefit of advanced fusion strategies is diminished. In such cases, simple or classical fusion methods can already achieve strong performance, and the room for improvement is limited.
>
> On the other hand, under a high missing rate, the available modalities for each sample are uncertain, significantly exacerbating modality imbalance and amplifying the differences in modality heterogeneity, which ultimately leads to modality training collapse. Our method effectively addresses the issue of modality heterogeneity under high missing rates by establishing the goal of maximizing mutual information between modal-specific embedding codes and semantic information.
>
> **For W4 and Q4, more comparison methods and experiments on other datasets.**
>
> Thank you for your constructive suggestions! In the revised manuscript, we added comparison experiments with the MMTFN, DSIP, and ShareGAN. However, MFASN relies on ROI-based feature extraction with non-uniform brain parcellation standards, differing fundamentally from the 3D image-based methods in our study and baselines. Therefore, we added additional experiments on the other three suggested methods.
>
> Since the above three methods can only handle two image modalities, MRI and PET, we only use the MRI and PET modalities to conduct the comparative experiment. Experimental results on AD vs. CN are shown below (The best results are bolded and the second-best results are italicized):
>
> | Missing| Metric\| | Ours   | ShareGAN | DSIP | MMTFN  |
> |-|-|-|-|-|-|
> | 0.1     | AUC   | **0.811**  | *0.790*    | 0.793      | 0.735  |
> |         | ACC   | **0.735**  | *0.723*    | 0.725      | 0.655  |
> |         | F1    | **0.749**  | *0.721*    | 0.715      | 0.642  |
> |         | SEN   | **0.762**  | 0.690    | *0.695*      | 0.680  |
> |         | SPE   | *0.714*  | **0.761**    | 0.755      | 0.631  |
>
> In addition, we perform comparison experiments on the new publicly available dataset **AIBL**.
>
> | Task | Missing | Metric | LMVCAT | Adapted | ShaSpec | GMD | MMTFN | ShareGAN | DSIP | Ours |
> |------|---------|--------|--------|----------------|---------|-----|-------|----------|------|------|
> | ADCN | 0.1     | AUC    | 0.918  | 0.919          | *0.920* | 0.912 | 0.890 | 0.845    | 0.845 | **0.936** |
> |      |         | ACC    | *0.871*  | 0.861          | 0.714 | 0.806 | 0.746 | 0.653    | 0.653 | **0.868** |
> |      |         | F1     | *0.906*  | 0.900          | 0.755 | 0.881 | 0.785 | 0.678    | 0.678 | **0.910** |
> |      |         | SEN    | *0.943*  | 0.872          | 0.783 | **0.960** | 0.806 | 0.761    | 0.761 | 0.915 |
> |      |         | SPE    | 0.677  | **0.831**          | 0.523 | 0.393 | 0.585 | 0.354    | 0.354 | *0.742* |
>
> **Due to space limitations**, we only show some of the results in the above tables. It can be seen from the tables that our method has achieved leading performance both in the two modalities and on the new AIBL dataset, especially in the key metrics AUC, ACC and F1.  Specifically, although SEN and SPE can often exhibit an imbalance—one metric being very high and the other very low due to their definitions in binary classification problems—our method still demonstrates competitive performance when considering these two metrics jointly.
>
> Please refer to more responses in the following comment block.

---

> > ### Comment · Reviewer_Pe76 · 2025-08-06
> >
> > Thank your for your detailed rebuttal! Although I still have some doubts about method innovation and experimental comparison, I think the current rebuttal has addressed most of my concerns. If supplementary experiments are added to the paper, I will update my score.

---

> > > ### Author Response · Authors · 2025-08-06
> > > **Thank you again for your feedback!**
> > >
> > > We sincerely thank you for your professional and constructive comments. We are further clarifying key issues based on the reviewers' comments, and the new supplementary experiments will be reflected in the final version. The code will also be open source to increase transparency.

---

> ### Author Response · Authors · 2025-07-31
> **Following Response 1**
>
> **For W5, the t-SNE visualizations and claim of a “strong modality preference.”**
>
> Thanks for your valuable comment! We agree that t-SNE, as a nonlinear dimensionality reduction technique, is primarily designed for visualization rather than quantitative assessment, and that the absence of visible separation does not always correlate with lack of predictive value. Therefore, for each modality, we extracted the modality-specific features and fed them independently into the same classifier and measure the classification performance. We performed this analysis under two training schemes: (a) trained using only mid-fusion, and (b) trained using our proposed hybrid late-fusion strategy.
>
> | Metric |  MRI | PET | CSF | CAD | \|MRI | PET | CSF | CAD |
> |--------|----------------------|-----|-----|-----|------------------------|-----|-----|-----|
> |         |   Mid-               |Fusion|Only|       | \|Hybrid           |Late|-Fusion|       |
> | AUC    | 0.467                | 0.198 | 0.787 | 0.999 |\| 0.726                  | 0.877 | 0.893 | 0.999 |
> | ACC    | 0.221                | 0.153 | 0.762 | 0.992 |\| 0.763                  | 0.802 | 0.844 | 0.992 |
> | F1     | 0.363                | 0.213 | 0.681 | 0.982 |\| 0.279                  | 0.649 | 0.725 | 0.982 |
>
> Conventional mid-fusion method leads to insufficient training of MRI and PET modalities due to premature model overfitting to CSF and CAD modalities. Furthermore, in fused multimodal shared representations, the model tends to overweight the corresponding CSF and CAD modalities to reduce the overall loss. Instead, using our hybrid late-fusion approach, modality-specific features are forced to participate in the overall decision process, resulting in improved feature discrimination of MRI and PET during training.
>
> **For W6, the clarity and consistency.**
>
> Thank you for your professional comments!
>
> (1) *For clarification of "spatial discontinuity caused by voxel serialization"*. We agree that the mention of "spatial discontinuity caused by voxel serialization" in the abstract may lack sufficient context for readers unfamiliar with this technical challenge. To improve clarity, we have revised Section 2.2.1 to provide a brief and accessible explanation:
>
>     "When converting 3D medical images into 1D sequences for state space modeling, traditional serialization methods (such as 3-axis or cross scanning) may break the natural spatial adjacency of voxels in 3D space, separating spatially adjacent voxels in a serialized arrangement. This disrupts local structural information, making it difficult for the model to capture both local and long-range spatial dependencies—a problem named 'spatial discontinuity.'"
>
> (2) *Organization about HSFE and HMamba Modules.*
> Thank you for your valuable comments regarding the organization of Section Method. In our manuscript, the HSFE module represents an overall multi-level encoding architecture designed to process 3D imaging data in a hierarchical fashion. Although the HMamba module serves as the key building block within HSFE, the HMamba module is a crucial innovation in our work. Given its technical novelty and central role, we chose to first present a detailed, stand-alone introduction to HMamba before describing how it is integrated within the hierarchical HSFE framework.
>
> We sincerely appreciate your comment, which made us realize that this structure could make it difficult for readers to grasp the overall framework. Therefore, we have added introduction at the opening of the Section Method:
>
>     "In this section, we present our proposed HAD framework in three parts: HMamba, HSFE, and the multimodal semantic representation learning framework. HMamba is the key building block for modeling long-range spatial dependencies in 3D medical images. HSFE is a multi-level feature extraction module constructed using HMamba blocks. The overall HAD framework consists of the HSFE module for 3D image encoding and the multimodal semantic representation learning framework for robust diagnosis under incomplete modalities"

---

> ### Author Response · Authors · 2025-07-31
> **Following Response 2**
>
> **For Q1, related work and comparison experiments.**
>
> Thank you for your valuable suggestions regarding the review of related work and the more comparative baselines. In the revised manuscript, we have provided a more comprehensive summary of recent advances in multimodal AD diagnosis. The updated related work now covers a broad range of representative methods, including but not limited to [1-3]:
>
> [1] Multi-modal deep learning model for auxiliary diagnosis of Alzheimer's disease, NeurCom.
> [2] Explainable deep-learning-based diagnosis of Alzheimer's disease using multimodal input fusion of PET and MRI Images, JMBE.
> [3] Multi-scale multimodal deep learning framework for Alzheimer's disease diagnosis, Computers in Biology and Medicine.
>
> We would like to highlight that most existing multimodal AD diagnostic methods focus on image modalities such as MRI and PET, or rely on single-modality data from private datasets. There are very few methods that can simultaneously handle the full combination of MRI, PET, CSF, and CAD, and are robust to missing modalities as required by our setting. Therefore, in our manuscript, we have included both medical-specific and general incomplete multimodal learning baselines for comparison. Following your suggestion, we have added more recent and strong baselines to our experimental results, and expanded our evaluations to include new dataset **AIBL**. These additions further demonstrate the effectiveness and robustness of our proposed method.
>
> **For Q2, data leakage in five-fold cross-validation.**
>
> Thank you for raising this important point regarding data leakage. In our experiments, we have taken the following measures to prevent any potential data leakage:
>
> 1.	Before performing cross-validation, we carefully ensured that no redundant or duplicate samples (including subjects with multiple scans or related entries) appear in both the training and testing splits. This was achieved by identifying unique subject IDs and assigning all scans from the same subject to a single fold only.
> 2.	All splits are subject-independent, and we have verified that no information from the test set is accessible during training or preprocessing.
>
> We have updated the manuscript to explicitly describe our redundancy removal and data splitting protocol for clarity and transparency.
>
> **For Q3, the modifications of baselines and more detailed implementation.**
>
> We fully acknowledge the importance of fair comparisons. However, existing multimodal AD diagnostic models that can both handle all four modalities and support missing modalities are extremely rare. Therefore, to enable a fair evaluation under our heterogeneous and incomplete modality setup, we made minimal and necessary modifications to some baseline architectures—such as adding modality-specific branches or modifying the prediction layers to accommodate the four modalities used in our paper. Importantly, these changes do not alter the core backbone or fundamental methodology of the compared models. We have clearly documented all such modifications in the revised manuscript to ensure transparency.
>
> We appreciate your suggestion regarding the need for more detailed training configurations. In the "Implementation Details" section of the revised manuscript, we will provide more comprehensive information to facilitate reproducibility:
>
>     “We implement our HAD using Python 3.9 and train the model with the Adam optimizer, setting an initial learning rate of 0.001. For all methods, we take the optimal result based on the validation set within 100 training epochs. All experiments are conducted on an Ubuntu 20.04 computing platform equipped with multiple NVIDIA RTX 4090 GPUs.”
>
> **For Q4, more datasets and the readability of the table.**
>
> Thank you for your meaningful comments! More experimental results are added (see above response **“For W4 and Q4”**). And in the revised manuscript, we have highlighted the best and second-best results to enhance the readability of the table.
>
> **Overall**: We admire your professional, meticulous and responsible attitude, and appreciate your constructive suggestions! Your valuable comments have greatly helped us improve the quality of our manuscript. We hope our response can address your concerns, and we are very happy to respond to your further questions.

---

### Official Review · Reviewer_KTeY · 2025-06-30

**Clarity:** 2
**Significance:** 2
**Originality:** 2
**Rating:** 4
**Confidence:** 4

**Summary:**

This paper proposes a novel hierarchical information aggregation framework (HAD) for incomplete multimodal diagnosis of Alzheimer's Disease (AD). This framework integrates heterogeneous data types, including MRI, PET, cerebrospinal fluid (CSF), and clinical assessment data (CAD)—even when some modalities are missing. Key innovations include a Hilbert curve-based Mamba block for efficient feature extraction, a hierarchical multi-scale feature module, and a mutual information-based objective to balance modality-specific and shared representations. Experiments on the ADNI dataset show that HAD outperforms recent baselines under various missing modality scenarios.

**Questions:**

- Q1: Can the authors include single-modality baseline results (i.e., models trained on each modality independently) to provide a clearer reference for the value of multimodal fusion?
- Q2: Is the random masking of modalities performed before or after the cross-validation split? Are multiple random seeds or runs used to ensure robust and reproducible results?
- Q3: Can the authors provide more details on the training and tuning procedures for the modified ResNet-50 baselines, and clarify whether these protocols are aligned with those of the proposed method?
- Q4: Can the authors provide additional experiments or analysis on how the number of HSFE hierarchical levels and the choice of scanning strategies in Mamba affect the model’s performance?
- Q5: In Tables 3–5, please consider highlighting the best results to improve the readability and transparency of the reported comparisons.

**Ethical Concerns:**

["NO or VERY MINOR ethics concerns only"]

**Final Justification:**

The authors’ rebuttal satisfactorily addressed my main concerns, especially regarding single-modality baselines, experimental protocols, and ablation studies. The additional clarifications and results improve the completeness and transparency of the work. While a few minor issues remain, they can be addressed in the camera-ready version and do not preclude acceptance. Considering the improvements and overall contribution, I am increasing my score to borderline accept.

**Limitations:**

yes

**Quality:**

3

**Strengths And Weaknesses:**

Strengths
- The proposed HAD framework combines multi-view Hilbert curve-based feature extraction, hierarchical modeling, and mutual information-based representation learning, which is technically sound and well-motivated.
- The methodology is generally well described, and the paper is well organized logically.
- The hybrid late-fusion and spatial continuity preservation ideas are innovative.

Weaknesses
- The paper does not include single-modality baselines, making it difficult to evaluate the true benefit of multimodal fusion over strong unimodal predictors.
- The procedure for random masking of modalities relative to the cross-validation split is not clearly described, raising concerns about experimental fairness and reproducibility.
- Many baselines are modified to use a ResNet-50 backbone, but there is little detail on their training protocols or alignment with the proposed method, which may affect the fairness of the comparisons.
- The impact of key architectural choices—such as the number of hierarchical levels in HSFE and the scanning strategy in Mamba—is not systematically explored through controlled experiments.

---

> ### Author Rebuttal · Authors · 2025-07-31
>
> We sincerely appreciate your encouraging comments on the technical soundness, methodological clarity, logical organization, and the innovative aspects of our proposed HAD framework. We also thank you for the insightful suggestions and comments, which are very helpful for us to improve the quality of the manuscript.
>
> **For W1 and Q1 about providing single-modality baselines:**
>
> Thank you for your suggestion to include single-modality experimental results. As requested, we have conducted additional experiments on the ADNI dataset (without missing modalities) for the MCI vs. CN classification task using each modality individually (MRI, PET, CAD, CSF), as well as our proposed HAD method. The detailed results are provided in the table below:
>
> | Metric  | MRI           | PET           | CAD           | CSF           | Ours (HAD)     |
> |-|-|-|-|-|-|
> | **AUC** | 0.731±0.037   | 0.702±0.024   | 0.946±0.013   | 0.648±0.036   | **0.960±0.013**    |
> | **ACC** | 0.700±0.026   | 0.658±0.015   | 0.873±0.013   | 0.616±0.054   | **0.903±0.016**    |
> | **F1**  | 0.759±0.038   | 0.723±0.038   | 0.878±0.015   | 0.671±0.070   | **0.917±0.016**   |
> | **SEN** | 0.819±0.069   | 0.750±0.124   | 0.879±0.019   | 0.689±0.115   | **0.925±0.027**    |
> | **SPE** | 0.528±0.065   | 0.528±0.177   | 0.868±0.042   | 0.508±0.145   | **0.874±0.023**    |
>
> The results demonstrate that each single modality provides valuable but limited predictive power. Our multimodal HAD framework consistently outperforms all single-modality baselines across on the key metric, highlighting the effectiveness and necessity of cross-modal fusion in our approach.
>
> **For W2 and Q2, random masks and cross-validation split.**
>
> Thank you for pointing out the ambiguity regarding the procedure for random masking of modalities relative to the cross-validation split. We apologize for any confusion caused by the previous description. The random modality masks are performed before the cross-validation split. To ensure experimental fairness and robustness, we use the same random seed for all methods when generating the random masks and splitting the dataset. We have revised the relevant section in the manuscript to clarify this procedure:
>
> “*Furthermore, in order to simulate different modality missing situations, we randomly mask [10%, 30%, 50%] instances of MRI, PET, and CAD modalities by filling in $0$ value at the missing position, while ensuring that at least one of the modalities is available for each subject. Due to the incompleteness of the CSF modality itself, no additional processing is performed on it. Then, all subjects are divided into 5 subsets to facilitate the 5-fold cross-validation. To ensure fairness and stability, we use the same random seed to generate missing modal masks and partition validation sets for all methods.*”
>
> **For W3 and Q3, modified ResNet-50 baselines and training and tuning details**
>
> Thank you for highlighting the need for more detailed information regarding the training and tuning procedures of the modified ResNet-50 baselines, as well as their alignment with our proposed method.
>
> **For modified ResNet-50 baselines.**
> Most existing dedicated multimodal AD diagnosis methods either lack support for arbitrary missing modalities, or are limited to imaging modalities only, without the ability to handle CAD and CSF data. General incomplete multimodal learning frameworks, on the other hand, primarily focus on modality fusion and typically lack shallow feature extractors specifically designed for our diverse multimodal data. Therefore, in order to ensure comprehensive and fair comparisons, we made necessary and minimal modifications to adapt existing approaches for the incomplete multimodal AD diagnosis scenario. For imaging modalities (MRI & PET), we use the widely adopted 3D ResNet-50 as the feature extractor. For non-image modalities (CSF & CAD), we use a simple MLP as the feature extractor. All modifications follow a simple and consistent principle, aiming to preserve each baseline’s original design while ensuring compatibility with our dataset. We further clarify the modifications for each baseline as follows:
>
> (1) **LMVCAT**: LMVCAT is a general-purpose missing-modality multi-label classification framework, but does not provide a 3D imaging feature extractor. We add a 3D ResNet-50 for shallow 3D feature extraction, followed by flattening to a feature vector. The multi-label loss is replaced by a single-label classification loss to fit our diagnostic task.
>
> (2) **ADAPTED**: ADAPTED is a plug-and-play compensation method for missing modalities. We integrate ADAPTED into the LMVCAT framework to evaluate its utility on our AD diagnosis task.
>
> (3) **DMRNet**: DMRNet is an incomplete multimodal representation learning framework originally designed for audio-visual tasks. To handle 3D MRI and PET, we use 3D ResNet-50 as the image feature extractor, consistent with the authors’ extensibility claims.
>
> (4) **ShaSpec**: ShaSpec is designed for missing-modality 3D medical image segmentation. To support CSF and CAD, we simply add two MLP branches for these modalities, without altering the core framework.
>
> (5) **GMD**: GMD is a missing multimodal learning framework, originally for audio/text input. We extend it with 3D ResNet-50 and MLP as feature extractors for the four modalities to handle our classification task.
>
> (6) **CM3T**: CM3T is a missing-modality AD diagnosis framework designed for the ADNI dataset, which is close to our setting, but lacks a PET branch. We replicate the MRI feature encoder for PET to enable fair comparison.
>
> (7) **TriMF**: TriMF is an incomplete multimodal fusion framework for medical data, using pairwise feature encoding for fusion. As its encoder only supports 2D images, we add a 3D ResNet-50 for MRI and PET feature extraction
>
> **For training and tuning details.**
> For all baseline methods and our proposed framework, we ensure that the training and hyperparameter tuning protocols are strictly aligned to guarantee fair and reproducible comparisons. Specifically, we replace or extend each baseline’s feature extractor as needed (using 3D ResNet-50 for MRI and PET, and MLP for CSF and CAD), and preprocess all input modalities with identical normalization and resizing procedures. For each baseline, we give priority to using the optimizer recommended in the original paper, and search for the optimal learning rate within the range of {1e^−5, 1e^−4, 1e^−3, 1e^−2,1e^−1} as well as the value suggested by the original work. Other hyperparameters for each method are set to the default values provided in the original paper or code; if default values are not available, we search for the optimal hyperparameters within the recommended ranges from the original work (where applicable). All models are trained for a maximum of 100 epochs. To ensure experimental consistency and fairness, we use the same random seed for all models when generating missing-modality masks and performing cross-validation splits, so that all methods are trained and evaluated on exactly the same training-validation partitions and missing data patterns. All these details are documented to facilitate reproducibility and transparency.
>
> **For Q4, more ablation experiments of the number of HSFE levels and scanning strategy**
>
> Thank you for your insightful suggestion regarding the analysis of HSFE hierarchical levels and scanning strategies in Mamba. In response, we have conducted additional experiments to systematically investigate:
>
> 1. Effect of HSFE Hierarchical Levels (k is the number of levels, conducted on MCI vs. CN with 50% missing rate)
>
> | Metric | k=1   | k=2   | k=3   | k=4   | k=5   |
> |-|-|-|-|-|-|
> | AUC   | 0.814±0.019   | 0.817±0.021   | 0.829±0.013   | **0.838±0.018**   | 0.834±0.014   |
> | ACC   | 0.726±0.026   | 0.722±0.017   | 0.732±0.021   | **0.732±0.028**   | 0.732±0.009   |
> | F1    | 0.734±0.022   | 0.723±0.022   | 0.728±0.029   | **0.740±0.032**   | 0.731±0.033   |
> | SEN   | 0.728±0.076   | 0.698±0.074   | 0.716±0.084   | **0.740±0.116**   | 0.706±0.106   |
> | SPE   | 0.732±0.091   | 0.756±0.079   | 0.748±0.092   | 0.735±0.120   | **0.763±0.121**|
>
> As shown in the table below, increasing the number of levels generally improves performance up to 4 levels, after which the benefit plateaus or slightly decreases, possibly due to overfitting or increased model complexity.
>
> 2. Effect of Scanning Strategies in Mamba
>
> | Method                | AUC   | ACC   | F1    | SEN   | SPE   |
> |-|-|-|-|-|-|
> | HMamba (ours)         | **0.838** | **0.732** | **0.740** | 0.740 | **0.735** |
> | HMamba w 3-axis scanning	     | 0.805 | 0.713 | 0.733 | **0.756** | 0.670 |
> | HMamba w cross scanning       | 0.800 | 0.722 | 0.723 | 0.720 | 0.725 |
>
> We have conducted explicit ablation studies comparing our multi-view Hilbert curve-based HMamba module with alternative voxel serialization strategies. The scanning strategy is replaced by the classical 3-axis scanning and cross scanning strategy for ablation experiments. Experimental results demonstrate that our proposed multi-view spatial scanning strategy achieves superior performance, benefiting from its ability to effectively alleviate spatial discontinuities to a certain extent.
>
> **For Q5, the readability of the table.**
>
> Thank you very much for your suggestion. In the revised manuscript, we have highlighted the best results in Tables 3–5 in bold to improve readability and transparency of the reported comparisons. This formatting allows readers to more easily identify the top-performing methods for each evaluation metric. We appreciate your feedback and have updated the tables accordingly.
>
> Thank you again for your serious and professional review comments. We believe that your suggestions will greatly help to improve the quality of our papers, and we hope that our response will address your concerns. If you have any other questions, we will be happy to discuss them with you!

---

> > ### Comment · Reviewer_KTeY · 2025-08-05
> >
> > Thank you for your detailed and constructive rebuttal. I appreciate the additional experiments, clarifications, and manuscript updates, which have satisfactorily addressed my main concerns regarding baseline completeness, experimental protocol, and ablation studies. I recognize the technical quality and contribution of your work. Provided that all promised changes and clarifications are reflected in the camera-ready version, I am willing to slightly increase my score.

---

> > > ### Author Response · Authors · 2025-08-05
> > > **Thank you again for your feedback and contrributions!**
> > >
> > > Thank you for your feedback and support. All clarifications and revisions will be reflected in the camera-ready version, and the code will also be made open source. Thank you again for your help in improving the quality of our manuscript.

---

### Official Review · Reviewer_d8G6 · 2025-07-01

**Clarity:** 3
**Significance:** 3
**Originality:** 3
**Rating:** 4
**Confidence:** 4

**Summary:**

This paper addresses the challenges of incomplete and heterogeneous multimodal data in Alzheimer’s disease diagnosis by proposing a unified diagnostic framework that can handle missing modalities and diverse data structures. The framework combines a multi-view HMamba module for efficient 3D image modeling, a hierarchical spatial feature extractor, and a mutual information–based semantic learning strategy to robustly integrate heterogeneous modalities.

**Questions:**

Q1. In this paper, the baseline models were adapted to fit the proposed tasks. Most of them appear to use ResNet-50 as the backbone. Could you elaborate on the rationale behind choosing ResNet-50 for these modifications?

Q2. The classification experiments are conducted across three binary tasks (e.g., AD vs. CN). Is there a specific reason for not including a 3-class classification setting (AD vs. MCI vs. CN), which could provide a more unified evaluation of the model’s overall capability?

Q3. In Figure 2, multiple multi-view sequences are shown. It is unclear whether all views are treated equally or if different weights are applied based on their importance. Does the model include any attention mechanism across views, or could this be a potential direction for improvement?

Q4. The proposed framework includes both the Hierarchical Spatial Feature Extractor (HSFE) and the HMamba module. However, no ablation studies are isolating the contribution of each module. If available, could you share any results that quantify the individual impact of these components on overall performance?

**Ethical Concerns:**

["NO or VERY MINOR ethics concerns only"]

**Final Justification:**

• The paper proposes a solid, well-structured, multimodal diagnostic framework for Alzheimer’s disease that can be used in real-world settings involving missing and heterogeneous modalities.
• Reviewers consistently found the approach to be technically sound, with comprehensive ablations and empirical performance.
• The key concern was the lack of sufficient theoretical or clinical justification for using Hilbert curve-based voxel serialisation, which plays a central role in the proposed architecture.
• While the rebuttal partially addressed this through comparisons and references, the benefit of the Hilbert-based design was not sufficiently highlighted or justified, leaving its necessity unclear.
• Despite the paper’s strengths in terms of its overall design and practical applicability, concerns remain about the justification of the Hilbert-based novelty, resulting in a borderline acceptance.

**Limitations:**

The authors appropriately address the main limitations of the study. No additional limitations were identified during my review.

**Paper Formatting Concerns:**

No major formatting issues were found.

**Quality:**

3

**Strengths And Weaknesses:**

**Strengths.**

This paper presents a clear and well-defined contribution by proposing a novel framework to address the challenges of incomplete and heterogeneous multimodal data in Alzheimer’s disease diagnosis. The proposed method is supported by solid mathematical formulations with each component clearly described and logically integrated. Furthermore, the paper includes extensive experiments to demonstrate the effectiveness of the approach and a well-organized appendix that provides additional implementation details and analyses to enhance clarity.

**Weaknesses.**

While the proposed HMamba module, which incorporates Hilbert curve–based sequences, is conceptually well-founded and architecturally coherent, there remain several aspects that warrant further clarification and refinement.

(1) First, although the use of Hilbert curves is conceptually motivated, the paper does not provide dedicated experiments or ablation studies to validate the effectiveness of this design choice, making it difficult to assess its contribution.

(2) Second, while the paper states that a reverse indexing operation is applied to map the serialized outputs $o^l$ back to the 3D voxel space, the mechanism is not clearly described through equations or architectural diagrams. As a result, it is unclear how the spatial alignment between $o^l$ and the voxel-wise weights $a^l$ is ensured, making the fusion process less transparent and potentially hindering reproducibility.

(3) Third, some of the figures and tables suffer from limited visual clarity, which may hinder reader comprehension. For example, in Figure 1, the directional flow of information is not clearly indicated due to ambiguous arrow styles. In Figure 3, the color distinction between 30% and 50% missing rates is not clearly distinguishable. It makes difficult to distinguish between conditions at a glance. Additionally, including SEN and SPE comparisons alongside ACC, AUC and F1 in Figure 3 would provide a more comprehensive evaluation. It would also improve readability if the best-performing models in Tables 3 to 5 were highlighted in bold.

This does not undermine the core idea of the paper, but suggest areas where clarity and reproducibility could be improved.

---

> ### Author Rebuttal · Authors · 2025-07-29
>
> Thank you for acknowledging the strengths of our study. We are grateful for your recognition and thoughtful feedback, which inspires us to continue our efforts. We will respond to the Weaknesses and Questions one by one.
>
> **For W1, ablation study of Hilbert curves.**
>
> *Response*: Thank you for your constructive feedback on the validation of the Hilbert curve design. We have conducted explicit ablation studies comparing our multi-view Hilbert curve-based HMamba module with alternative voxel serialization strategies, as shown in Table 1 (see Section 4.4):
> | Method                | AUC   | ACC   | F1    | SEN   | SPE   |
> |-|-|-|-|-|-|
> | HMamba (ours)         | **0.838** | **0.732** | **0.740** | 0.740 | **0.735** |
> | HMamba w 3-axis scanning     | 0.805 | 0.713 | 0.733 |**0.756**| 0.670 |
> | HMamba w cross scanning      | 0.800 | 0.722 | 0.723 | 0.720 | 0.725 |
>
> In our experiments, we replaced the scanning method in the classic visual Mamba to confirm the advancement of our multi-view Hilbert-based curve scanning method. We named them as “HMamba w SegMamba” and “HMamba w VMamba”, in which the scanning strategy is replaced by the classical 3-axis scanning [1] and cross scanning [2] (To avoid ambiguity, we renamed them in the revised manuscript as “HMamba w 3-axis scanning” and “HMamba w cross scanning”). Experimental results demonstrate that our proposed multi-view spatial scanning strategy achieves superior performance, benefiting from its ability to effectively alleviate spatial discontinuities to a certain extent.
>
> [1] Segmamba: Long-range sequential modeling mamba for 3d medical image segmentation, MICCAI, 2024.
>
> [2] Vmamba: Visual state space model, NuerIPS, 2024.
>
> **For W2, description of the inverse indexing operation**
>
> *Response*: Thank you for pointing out the need for a clearer and more detailed description of the reverse indexing operation and voxel-wise fusion. After applying the multi-view Hilbert curve-based serialization to the 3D voxel grid, each view produces a serialized 1D feature sequence $o^l \in \mathbb{R}^{d_i^3}$. To restore the spatial correspondence with the original 3D space, we apply a reverse mapping using the inverse Hilbert curve index:
>
> $o^l_{[h]}\stackrel{Index}{\longrightarrow} o^l_{[(x_c, y_c, z_c)]}$
>
> In practice, this is implemented as:
>
> ```
> # Pseudocode for reverse mapping
> # Once the order N is given, the sequence coordinates are uniquely determined
> # L is the length of the serialized feature
> for h in range(L):
>     x, y, z = hilbert_index(h, N)
>     o_cube[x, y, z] = o_serialized[h]
> ```
> We only need to store the index vector as a list during the serialization process. During the reverse serialization process, the three-dimensional coordinates saved at the corresponding positions of the sequence can be retrieved and used to index the voxel space. The detailed implementation process can be referred to in the supplementary materials.
>
> **For W3, the readability of figures and tables**
>
> *Response*:
> Thank you for your valuable suggestions.
>
> (1)	We have redrawn Figure 1 to improve the clarity of information flow. The arrows have been made more explicit and consistent to avoid any ambiguity regarding the direction or type of information transfer between modules. In addition, each method is now represented by clearly distinguishable colors.
>
> (2)	Due to space constraints in the main text, we have provided the complete experimental results—including Sensitivity (SEN) and Specificity (SPE) comparisons—alongside ACC, AUC, and F1 in Appendix A.6 as tables. This allows readers to comprehensively evaluate all relevant metrics. Besides, we have supplemented the line charts of the indicators SEN and SPE.
>
> (3)	In the revised manuscript, we have highlighted the best-performing results in bold in Tables 3 to 5, following the reviewer’s suggestion. This makes it easier for readers to quickly identify the most effective methods across different tasks and missing rates.
>
> **For Q1, using ResNet-50 backbone in baselines**
>
> *Response*: On the one hand, ResNet-50 is one of the most widely used and well-established visual feature extractors in medical imaging tasks, especially in AD diagnosis, such as [1-3]. Its strong representational power and proven performance across a range of datasets make it a popular and reliable choice for fair comparison. On the other hand, methods that can natively handle all four heterogeneous modalities (MRI, PET, CSF, and CAD) are rare. To ensure a fair and consistent comparison among all baseline methods, we replaced or added ResNet-50 as the backbone feature extractor for all methods requiring adaptation. By standardizing the visual feature extraction across baselines, we minimize the influence of backbone architecture on the final results, allowing a clearer assessment of each method’s multimodal fusion and handling of missing data.
>
> [1] 3D multimodal fusion network with disease-induced joint learning for early Alzheimer’s disease diagnosis, TMI, 2024.
>
> [2] Robust Multimodal Learning via Representation Decoupling, ECCV, 2024.
>
> [3] Multi-modal Learning with Missing Modality via Shared-Specific Feature Modelling, CVPR, 2023.
>
> **For Q2, 2-class vs. 3-class classification settings**
>
> Thank you for the insightful and professional question regarding our use of three binary classification tasks rather than a unified 3-class (AD vs. MCI vs. CN) setting. This is up to several scientific and clinical considerations:
>
> (1) *Class Boundary Ambiguity and Sample Distribution*. MCI represents a “transitional state” in the progression from CN to AD, and its clinical presentations and imaging features typically lie between those of AD and CN. Importantly, MCI can be further stratified into progressive MCI (pMCI, more similar to AD) and stable MCI (sMCI, more similar to CN), resulting in strong polarization and substantial overlap at both ends. This unstable MCI label leads to severe class boundary ambiguity and sample imbalance in 3-class settings, making it particularly challenging for models to distinguish all three categories with high precision [1,2].
>
> (2) *Clinical relevance and stepwise evaluation*. Binary classification tasks allow us to independently assess the model’s performance in early screening (MCI vs. CN), disease progression (AD vs. MCI), and disease identification (AD vs. CN). Each task emphasizes distinct features and scenarios, facilitating model interpretability and clinical translation [2,3]. This design also mirrors real-world diagnostic pathways: clinicians typically first rule out healthy controls, then further differentiate between mild impairment and overt disease, following a stepwise decision-making process. Notably, combining three binary classifiers via ensemble strategies (e.g., majority voting or probability fusion) can enhance overall recognition accuracy and adapt flexibly to different clinical objectives [4].
>
> (3) *Alignment with standard practice and existing work*. The vast majority of prior AD research focuses on binary classification settings, enabling direct and fair comparison with existing work. While unified 3-class classification is a valuable direction, we believe that focusing on the widely adopted binary paradigm is most appropriate for this study. We plan to explore multi-class classification in future work.
>
> [1] Multimodal Attention-based Deep Learning for Alzheimer’s Disease Diagnosis. 2022.
>
> [2] Machine Learning-Driven GLCM Analysis of Structural MRI for Alzheimer’s Disease Diagnosis, 2024.
>
> [3] Exploring 3D Transfer Learning CNN Models for Alzheimer's Disease Diagnosis from MRI Images, 2023.
>
> [4] Classification of Alzheimer’s Disease, Mild Cognitive Impairment, and Normal Controls With Subnetwork Selection and Graph Kernel Principal Component Analysis Based on Minimum Spanning Tree Brain Functional Network, 2018.
>
>
> **For Q3, multi-view dynamic fusion.**
>
> Thank you for your question about the multi-view dynamic fusion. Our model does not treat all views equally. Instead, we propose a dynamic multi-view fusion module, which learns voxel-wise attention weights across views. This allows the model to adaptively focus on more informative views for each spatial location. The fusion process is formulated as:
>
> $ \hat{o} = \sum_{l=1}^{L} a^l \odot \tilde{o}^l $
>
> The attention weights $a^l$ for $l$-th view are learned automatically during training, ensuring each voxel aggregates information from different views according to their importance. This mechanism is described in detail in our manuscript as “multi-view dynamic fusion”.
>
> **For Q4, ablation experiments of HSFE and HMamba module.**
>
> *Response*: Thank you for your valuable suggestion. In our framework, HMamba is a core sub-module within HSFE, specifically designed for long-range voxel-wise feature extraction. To explicitly quantify the contribution of each component, we have supplemented the following ablation experiments, in which we assessed the performance of the following model variants:
>
> | Method | AUC | ACC | F1 | SEN | SPE |
> |-|-|-|-|-|-|
> | HSFE w/o HMamba |0.816 | 0.713 | 0.707 | 0.690 | 0.738|
> | HAD w/o HSFE | 0.813 | 0.710 | 0.706 | 0.675 | **0.757** |
> | **HAD**| **0.838** | **0.732** | **0.740** | **0.740** | 0.735 |
>
> In this variant, we replace the HMamba block inside HSFE with a 3D ResNet50 module, so that the hierarchical structure is preserved but the long-range modeling of HMamba is removed. Then, we remove the entire HSFE structure and use a vanilla 3D ResNet50 as the backbone for the image modality. All the experiments are conducted on the task of MCI vs. CN with 50% missing rate.
> These results demonstrate that both HSFE and HMamba independently improve performance, and their combination achieves the best results.
>
> We have made targeted revisions to the manuscript based on your suggestions, and we hope that the above responses will address your concerns well.

---

> > ### Comment · Reviewer_d8G6 · 2025-08-05
> >
> > Thank you to the authors for their thoughtful and detailed rebuttal.
> >
> > I appreciate the additional ablation study in Table 1 comparing different scanning methods (e.g., 3-axis, cross scan), which provides helpful empirical evidence. However, the performance improvements from using Hilbert curves are relatively modest and not accompanied by standard deviation analysis. Moreover, the justification for selecting Hilbert curves over other locality-preserving alternatives remains largely empirical, without strong theoretical or task-specific motivation.
> >
> > Given that Hilbert curve–based serialization is one of the paper’s key innovations, a more compelling rationale—through theoretical analysis, visualization of spatial continuity, or clinical relevance—would significantly strengthen the contribution.
> >
> > Overall, I maintain a borderline accept rating. While the proposed framework is technically sound, further clarification on this design choice would help solidify the novelty and impact of the work.

---

> > > ### Author Response · Authors · 2025-08-05
> > > **Thanks for your feedback!**
> > >
> > > Thank you for your thoughtful comments and suggestions regarding the justification for Hilbert curve–based serialization.
> > >
> > > The primary motivation for adopting the Hilbert curve is its superior **locality-preserving properties** compared to other common scanning methods such as raster (3-axis) or Morton (Z-order) curves. Some studies have shown that the Hilbert curve more effectively maps multidimensional data to one dimension while minimizing the likelihood that spatially adjacent points are mapped far apart in the serialized sequence.
> > >
> > > For example, Moon et al. [1] and Haverkort & van Walderveen [2] rigorously analyzed the clustering and locality-preserving characteristics of various space-filling curves. Their findings indicate that the Hilbert curve consistently achieves lower "dilation" and "clustering number"—metrics that quantify the preservation of local neighborhoods—than alternatives like Morton or raster scans, especially in 3D settings. This means that, for a fixed window size, points that are close in the original spatial domain tend to remain close in the serialized representation, which is beneficial for downstream tasks such as neural network modeling and local feature aggregation.
> > >
> > > In addition to the quantitative results, we have provided the visualizations in the appendix to further illustrate the spatial continuity and scanning behavior of the 3D Hilbert curve. Specifically, Figure 5 presents schematic diagrams of 3D Hilbert curves with different orders, while Figure 6 shows multi-view visualizations of the 3D Hilbert curve. We believe these visualizations effectively demonstrate how the Hilbert curve traverses the 3D volume and preserves spatial locality from various perspectives.
> > >
> > > Furthermore, while prior works have demonstrated the effectiveness of the Hilbert curve in preserving spatial locality, we recognize that voxel serialization inevitably introduces some discontinuities, and even the Hilbert curve cannot fully eliminate this issue. To address this, our work proposes a novel multi-view Hilbert curve approach: by applying Hilbert curves from multiple orientations, we enhance the coverage of spatially adjacent voxels from different directions. Furthermore, our dynamic fusion strategy aggregates the information from these multiple scanning perspectives, further mitigating the limitations of single-directional serialization and improving continuity in the aggregated representation. We believe this is a key innovation of our method.
> > >
> > > **Overall**, we sincerely thank you for highlighting the need for a stronger theoretical motivation behind our choice of the Hilbert curve. In our revised manuscript, we will further elaborate on this motivation and include appropriate citations to established findings regarding the locality-preserving properties of the Hilbert curve. We hope this would address your concerns and will be happy to respond to further questions.
> > >
> > > [1] Analysis of the clustering properties of the Hilbert space-filling curve, TKDE.
> > > [2] Locality-preserving properties of space-filling curves, Computational Geometry.

---

### Author Response · Authors · 2025-08-03
**Thank you to all reviewers and AC.**

We sincerely appreciate the time and effort invested by the reviewers in evaluating our manuscript, as well as their constructive and encouraging feedback. We are grateful for their recognition of our work’s contributions and the thoughtful comments that have helped us improve the paper.

Overall, according to the reviews, our summarization is as follows:

**Strengths:**

* The paper proposes a novel framework for Alzheimer’s disease diagnosis that integrates multi-view Hilbert curve-based feature extraction, hierarchical modeling, and mutual information-based learning, and introduces innovative ideas such as hybrid late-fusion and spatial continuity preservation. (Reviewer d8G6, KTeY, Pe76, S54Q)

* The manuscript is generally well organized and clearly written, with detailed methodological descriptions, extensive experiments, and a well-structured appendix providing additional implementation details and analyses, making the study accessible and easy to follow. (Reviewer d8G6, KTeY, S54Q)

* The work addresses the practically significant problem of incomplete and heterogeneous multimodal data in Alzheimer’s disease diagnosis, which is highly relevant for real-world clinical applications. (Reviewer Pe76)

**Weaknesses:**

* **Experiments:** Some experimental results are insufficient, such as the lack of single-modality baselines, limited comparison with recent state-of-the-art methods, absence of model complexity analysis, and incomplete exploration of key architectural choices (e.g., ablation for Hilbert curve, hierarchical levels, scanning strategies).

* **Presentation:** There are issues affecting clarity and reproducibility, including unclear or insufficient explanation of some technical details, missing or confusing information about experimental protocols, lack of citation to relevant literature, and occasional inconsistencies or ambiguities in terminology, figures, and tables.

Based on the reviewers’ valuable suggestions and comments, we have conducted additional experiments and enhanced the clarity of the manuscript as requested. We believe these improvements have significantly strengthened the quality of our work.

Thank you again for your professional and careful review comments. Your encouragement and suggestions motivate us to continue contributing to the community.


If you have any further questions or concerns, we would be happy to address them. Your feedback will be highly appreciated.

---

### Note · Authors · 2025-08-13

We would like to sincerely thank all reviewers and the AC for the thoughtful comments and constructive feedback during the discussion phase. All reviewers acknowledged the technical quality, clarification, and contribution of our work. All reviewers confirmed that our rebuttal has addressed all or most of their concerns, and consistently suggested positive ratings for our manuscripts.

We would like to thank everyone once again for their efforts in improving the quality of our manuscripts. All updates will be presented in the final version.

---

### Decision · Program_Chairs · 2025-09-17

**Decision:**

Accept (poster)

**Comment:**

This paper proposes a heterogeneous multimodal diagnosis framework for Alzheimer's Disease which can handle arbitrary incomplete modalities by combining multi-view Hilbert curve-based feature extraction, hierarchical modeling, and mutual information-based representation learning.

Four reviewers, each with confidence level 4, have given scores of 4, 4, 4 (borderline accept) and 5 (accept). Reviewers agree on well motivated algorithmic development and empirical results that show superiority over state-of-the-art alternatives. Several concerns were raised around additional empirical results, e.g., ablations that demonstrate key contributions of Hilbert curves in their method, comparisons that show modality-wise performance, running time comparisons. Most of these questions were satisfactorily addressed by the authors in their rebuttal.

With all the additional results and discussion incorporated, I believe the paper presents a strong contribution in an important application domain. I strongly suggest adding standard errors to the ablation study (as suggested by a reviewer) as well as to all other comparative analyses, wherever applicable.